# AMemGym: Interactive Memory Benchmarking for Assistants in Long-horizon Conversations

**Cheng Jiayang**[1,2*], **Dongyu Ru**[2*], **Lin Qiu**[2†], **Yiyang Li**[2]
**Xuezhi Cao**[2], **Yangqiu Song**[1], **Xunliang Cai**[2]
[1]The Hong Kong University of Science and Technology    [2]Meituan
jchengaj@cse.ust.hk, {rudongyu, qiulin07}@meituan.com

## Abstract

Long-horizon interactions between users and LLM-based assistants necessitate effective memory management, yet current approaches face challenges in training and evaluation of memory. Existing memory benchmarks rely on static, off-policy data as context, limiting evaluation reliability and scalability. To address these gaps, we introduce AMemGym, an interactive environment enabling on-policy evaluation and optimization for memory-driven personalization. AMem-Gym employs structured data sampling to predefine user profiles, state-dependent questions, and state evolution trajectories, enabling cost-effective generation of high-quality, evaluation-aligned interactions. LLM-simulated users expose latent states through role-play while maintaining structured state consistency. Comprehensive metrics based on structured data guide both assessment and optimization of assistants. Extensive experiments reveal performance gaps in existing memory systems (e.g., RAG, long-context LLMs, and agentic memory) and corresponding reasons. AMemGym not only enables effective selection among competing approaches but also can potentially drive the self-evolution of memory management strategies. By bridging structured state evolution with free-form interactions, our framework provides a scalable, diagnostically rich environment for advancing memory capabilities in conversational agents.[1]

## 1 Introduction

A crucial objective in the development of assistants based on Large Language Models (LLMs) is to achieve long-horizon conversational capabilities—that is, the ability to effectively organize, manage, and utilize memory across extended sequences of dialogue turns. Robust memory management forms the foundation for fulfilling complex user requests, tailoring responses to users' latest implicit states, and personalizing suggestions and recommendations based on interaction history. However, progress in advancing conversational memory systems for assistants is hampered by a critical bottleneck that affects both scalable training and reliable evaluation: the data used in existing benchmarks.

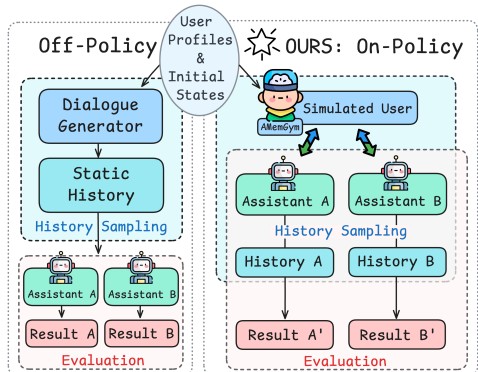

Figure 1: On-policy v.s. off-policy evaluation for assistants' memory.

Current benchmarks typically rely on static, off-policy data for evaluation (Xu et al., 2022; Wu et al., 2024; Hu et al., 2025), rather than on-policy interactions. Figure 1 illustrates the distinction between

---

*Equal contribution. Work done during an internship at Meituan.
†Project Lead.
[1]https://agi-eval-official.github.io/amemgym/

Table 1: A comparison of features across agent memory benchmarks.

| Benchmark | Eval. Mode | Optim. Feedback | Automation Level | Context Length | Eval. Metrics |
|---|---|---|---|---|---|
| MSC (Xu et al., 2022) | Static | ✗ | Manual | 1.2K | - |
| RealTalk (Lee et al., 2025) | Static | ✗ | Manual | 17K | Emotional Intelligence, Persona Simulation, Memory Probing (F1, accuracy) |
| DialSim (Kim et al., 2024) | Static | ✗ | Manual | - | QA Accuracy |
| LoCoMo (Maharana et al., 2024) | Static | ✗ | Semi-Automated | 9.2K | QA Accuracy, Summarization, Generation |
| PerLTQA (Du et al., 2024) | Static | ✗ | Semi-Automated | - | QA Accuracy |
| LongMemEval (Wu et al., 2024) | Static | ✗ | Semi-Automated | Configurable (115K, 1.5M) | Retrieval Recall, QA Accuracy |
| PersonaMem (Jiang et al., 2025) | Static | ✗ | Fully Automated | Configurable (32K, 128K, 1M) | QA Accuracy |
| AMEMGYM (This Work) | Interactive | ✓ | Fully Automated | Configurable | Overall (Accuracy, Normalized Memory Score) and Diagnosis (Write, Read, Utilization). |

the two approaches. Off-policy evaluation, in which an assistant is tested on conversational data that it did not produce during actual interactions, presents several fundamental limitations. First, it fails to capture the assistant's true interactive nature, as the evaluation data does not reflect the consequences of the assistant's own conversational choices—a critical issue for evaluation realism. Second, because the evaluation is biased, memory optimization could be misdirected. Finally, the manual curation of these evaluation scenarios (Lee et al., 2025; Kim et al., 2024) is costly and does not scale for comprehensive testing across diverse, long-horizon conversational contexts.

To enable on-policy evaluation and provide reliable feedback for optimization, it is essential to employ a simulated user that can strategically reveal information and pose relevant questions, a technique that has demonstrated promise in other domains such as tool use (Wang et al., 2023; Lu et al., 2025). However, deploying simulated users in open-ended conversational environments presents unique challenges. These include determining what information to disclose dynamically while maintaining a natural and coherent dialogue, as well as ensuring the generation of diverse, high-quality data that remains sufficiently controlled for reliable evaluation.

To address these gaps, we introduce AMEMGYM, an interactive environment designed for the on-policy evaluation and optimization of memory in long-horizon conversations. AMEMGYM grounds free-form interactions in structured data generated through a schema-based approach. The framework predefines user profiles, state-dependent questions, and state evolution trajectories to enable the cost-effective generation of high-quality interactions aligned with evaluation targets. LLM-simulated users then expose these latent states through natural role-play, ensuring consistency with the structured state evolution. Periodic evaluation during interactions, using both overall and diagnostic metrics, guides assessment and optimization of memory capabilities. Our contributions are threefold:

1. We introduce AMEMGYM, a novel framework for the on-policy evaluation of conversational memory. By grounding free-form interactions in a structured state evolution, AMEMGYM creates a scalable and diagnostically rich environment to reliably assess and advance the memory capabilities of conversational agents.

2. We empirically demonstrate the reuse bias and potential drawbacks of off-policy evaluation, and conduct the first **extensive on-policy evaluation** of popular memory systems. Our results highlight the reliability of AMEMGYM for evaluating memory in the context of personalization.

3. We provide a proof of concept for **agent self-evolution**, showing that an agent can use environmental feedback within AMEMGYM to autonomously refine its memory management policy.

## 2 RELATED WORK

**Benchmarks for agent memory evaluation.** The evaluation of agent memory has progressed from long-context, single-turn tasks like the needle-in-a-haystack (NIAH) test and NoLiMa (Modarressi et al., 2025) to more realistic multi-turn conversational datasets such as Multi-Session Chat (MSC) (Xu et al., 2022), RealTalk (Lee et al., 2025), and DialSim (Kim et al., 2024). While these introduced more authentic dialogue patterns, their reliance on manual curation limited their scale and diversity. To address this, automated data generation frameworks like LoCoMo (Maharana et al., 2024), PerLTQA (Du et al., 2024), LongMemEval (Wu et al., 2024), PersonaMem (Jiang et al., 2025), and MemoryAgentBench (Hu et al., 2025) were developed. However, a critical limitation unites nearly all existing benchmarks: they rely on static, off-policy data (Table 1). This

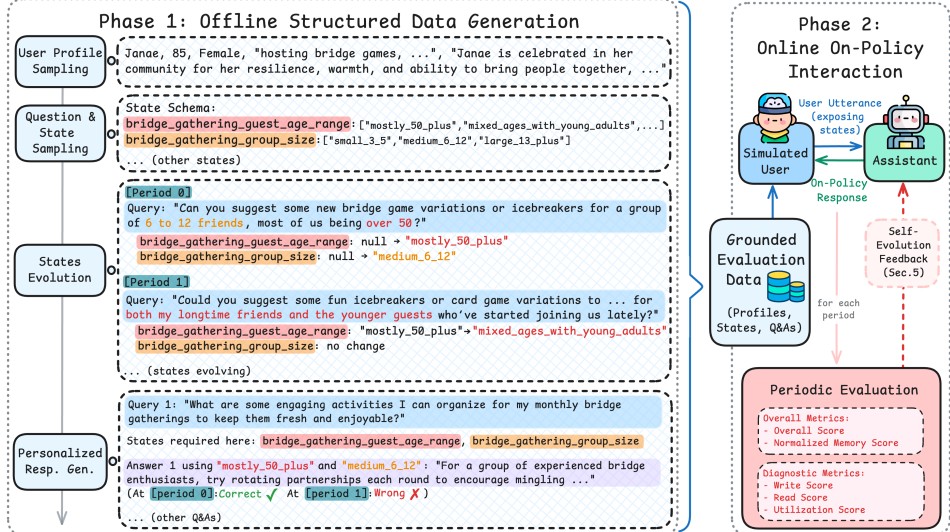

Figure 2: An overview of the AMemGym framework.

approach fails to capture an agent's true interactive performance, as the evaluation data does not reflect the consequences of the agent's own actions, misleading optimization.

**Interactive agent evaluation by user simulation.** An alternative line of research has focused on interactive, on-policy evaluation environments that employ user simulators. This approach has proven effective in domains like tool-use, where simulators provide robust on-policy evaluation (Wang et al., 2023; Lu et al., 2025). Similarly, efforts like CollabLLM (Wu et al., 2025) have successfully employed user simulation to train models for improved long-term collaboration. Applying this interactive paradigm to memory evaluation, however, introduces unique challenges: a simulator must strategically reveal information over a long-horizon conversation while maintaining a natural flow and generating interactions that are both diverse and controlled enough for reliable assessment. AMEMGYM directly addresses these challenges by introducing a schema-based approach that grounds free-form, LLM-driven role-play in a structured state evolution plan, which enables the controlled and scalable generation of on-policy, memory-focused evaluation scenarios.

## 3 AMEMGYM

AMEMGYM provides an interactive environment for benchmarking and optimizing personal assistant memory, with the scenario and the task described below.

**LLM-based Assistants.** An LLM-based assistant takes as input the observation (user input) $o_t$ and provides output responses $a_t$ (a sequence of tokens) based on its policy $\pi$ and its internal memory at that time $m_t$ (e.g., tokens in the context window, text snippets written to an external index, or its own parameters): $o_t, m_t \xrightarrow{\pi} a_t, m_{t+1}$. The internal memory is updated through interactions.

**Personalization with Memory.** To effectively serve users with dynamically evolving personal states, assistants described above must continuously track user states through interaction histories $\tau_t = [o_0, a_0, o_1, a_1, \ldots, o_t]$ and deliver responses optimized for their latest latent states captured by $m_t$. In reality, the length of $\tau_t$ often goes well beyond the optimal context length of most LLMs. Therefore, an effective information compression or memory mechanism is crucial for assistants to maintain accurate and up-to-date user modeling. In this context, *states* refer to comprehensive personal information crucial for enabling the intelligent assistant to sustain meaningful conversations and address user-relevant concerns. This includes user preferences, habits, plans, and environmental conditions, among other factors.

An overview of our framework[2] is presented in Figure 2. We begin by describing the structured data sampling process that forms the foundation of our evaluation framework (§ 3.1), then detail how on-policy interactions are generated with grounded structured data (§ 3.2). We present comprehensive evaluation metrics that assess both overall memory performance and provide diagnosis for different

---

[2]We use gpt-4.1 (OpenAI, 2025a) for structured data generation and user simulation.

memory operations (§ 3.3). Finally, we provide meta-evaluation results to show reliability of the fully-automated process (§ 3.4).

## 3.1 STRUCTURED DATA GENERATION FOR ON-POLICY INTERACTION

Evaluating memory is challenging due to the high cost of verifying correctness in long, noisy conversations. To address this, we use a reverse-engineering strategy: starting from target evaluation questions, we trace back to identify key user state variables for personalization, their possible temporal changes for a simulated user, and the personalized responses for each experienced state combination. This serves as a structured foundation that enables grounded interactions and automatic evaluation. Detailed prompts for each sampling step are provided in Appendix C.3.

**User Profile Sampling.** We begin by sampling user profiles that serve as the contextual backbone for subsequent steps. For broad domain coverage, we use 100K personas from Nemotron-Personas (Meyer & Corneil, 2025) as the pool. Custom sampling strategies can be easily applied for specific applications to better accommodate target real-world distributions.

**Question Sampling.** The process starts with a user profile, $p$, used to sample a set of evaluation questions, $\mathcal{Q}_p$. For each question $q_i \in \mathcal{Q}_p$, an LLM extracts the information types required for a personalized answer. These types $\mathcal{S}'_i$ are occasionally redundant across questions (e.g., "experience_level" and "years_of_work"). Therefore, they are merged and refined by an LLM into a canonical **global state schema**, $\Sigma = \bigcup_i \mathcal{S}'_i$. The schema defines a set of $M$ unique state variables $(s_j)$ and their possible discrete values set $(V_j)$: $\Sigma = \{(s_j, V_j)\}_{j=1}^{M}$. This comprehensive schema serves as the complete set of trackable user states for the entire simulation.

**User States Evolution.** We then simulate a realistic progression of the user's states over $N_p$ periods. The state at the end of each period $t$ is captured by a **state vector**, $\sigma_t$, a full assignment where each variable $s_j$ is given a value $v_j$ from its corresponding set of possibilities $V_j$: $\sigma_t = \{(s_j, v_j) \mid (s_j, V_j) \in \Sigma\}$. Each state transition is prompted by a narrative **life event**, $e_t$, providing context for the change ($\sigma_{t-1} \xrightarrow{e_t} \sigma_t$). The resulting **state evolution trajectory**, $\mathcal{T}_\sigma = (\sigma_0, \ldots, \sigma_{N_p})$, provides the ground-truth for the user's state throughout the simulation.

To create the inputs for on-policy interaction in each session, we generate a series of natural language utterances that the simulated user will say initially. Within each period $t$, an utterance $u_{t,k}$ is designed to implicitly *expose* a small related subset of the user's current state, $\sigma_{\text{exposed}} \subset \sigma_t$. This is generated by a function $G_{\text{utt}}$ conditioned on the states to be revealed and the user's profile: $u_{t,k} = G_{\text{utt}}(\sigma_{\text{exposed}}, p)$. These pre-generated, state-bearing utterances form a core part of the structured data blueprint. They are used to initiate conversational turns during the on-policy interaction phase (§ 3.2).

**Personalized Response Generation.** Finally, to create the evaluation ground truth, we generate personalized answers for each predefined question $q_i$. Each question requires a subset of state variables, $\mathcal{S}_{\text{req}}(q_i) \subset \{s_1, \ldots, s_M\}$, and a specific assignment of values to these variables is a **state variant**, $\nu$: $\nu = \{(s_j, v_j) \mid s_j \in \mathcal{S}_{\text{req}}(q_i), v_j \in V_j\}$. For each pair $(q_i, \nu)$, we generate a distinct answer $r_{i,\nu}$. To ensure a high-quality, one-to-one mapping, a reflection step verifies that the answer is unambiguous: it is accepted only if an LLM classifier $C$ can recover the variant from the question-answer pair, i.e., $C(q_i, r_{i,\nu}) = \nu$.

## 3.2 ON-POLICY INTERACTION

Different from prior static evaluation on long-context LLMs or memory agents (Xu et al., 2022; Maharana et al., 2024; Wu et al., 2024; Jiang et al., 2025), we sample on-policy interactions as in Figure 1. Given the offline structured data sampled in Section 3.1, our user simulator interacts with the target assistant to expose this information through natural conversation. This step outputs a (possibly long-context) dialogue history $\tau$. Later in Section 4.2, we demonstrate the necessity of on-policy evaluation.

**State Exposure.** To enable reliable evaluation, key user states—those that change between periods—must be clearly reflected in the conversation history. This is achieved by using the grounded utterances ($u_{t,k}$) that were pre-generated as part of the structured data. For benchmarking con-

sistency, we use these fixed initial state-bearing utterances to begin each conversational session, ensuring that the necessary information is introduced into the dialogue.

**Role-Play with LLMs.** Conversation generation is performed by a user LLM, which role-plays based on the user profile and state evolution. It is configured with: (1) a system prompt template incorporating the user profile, (2) current states $\sigma_t$, and (3) the latest conversation context. The user LLM produces responses conditioned on dialogue history and underlying states, ensuring coherent alignment between free-form conversation and structured state evolution.

## 3.3 EVALUATION METRICS

Given the grounded interactive environment, assistants are prompted to answer all evaluation questions after each interaction period. These responses provide feedback for agent builders to assess and optimize assistants (§ 4), and enable assistants to self-improve (§ 5), based on the evaluation metrics described below.

**Overall Evaluation.** We use the average question answering accuracy as the metric for evaluating end-to-end performance on our benchmark, denoted as the *overall* score. This metric captures the model's ability to integrate both personalization (tailoring responses based on specific user states) and memory (retaining user states from previous conversations) to achieve high performance. To provide a clearer view on memory, we introduce normalized *memory* scores. It isolates the memory component from raw task performance by normalizing the overall accuracy between a random baseline (lower bound) and an upper bound (UB) with perfect memory access. For each evaluation period, the score is computed as: $S_{\text{memory}} = \frac{S_{\text{overall}} - S_{\text{random}}}{S_{\text{UB}} - S_{\text{random}}}$. The upper bound $S_{\text{UB}}$ is determined by providing the assistant with ground-truth user states at evaluation time, thereby entirely bypassing the memory retrieval process. It measures the assistant's reasoning and application capabilities when required information is perfectly available.

**Diagnostic Evaluation.** We decompose failures in overall question answering into three distinct operational stages of memory processing: *write*, *read*, and *utilization*. Corresponding failure rates enable systematic error attribution. For each user state, we query its value at every evaluation period. If the assistant demonstrates knowledge of all relevant state values but still fails to answer an overall evaluation question correctly, we classify this as a *utilization* failure.

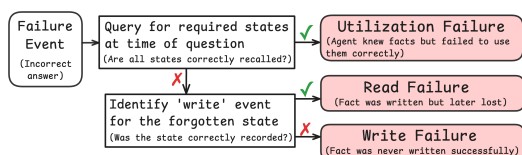

Figure 3: An overview of diagnostic metrics: *write, read,* and *utilization*.

Otherwise, we examine the state query results at the nearest write position to distinguish between *write* and *read* failures (Figure 3).

## 3.4 META-EVALUATION

To validate the data quality of AMEMGYM, we conducted a three-stage meta-evaluation with human annotators. First, we assessed *state exposure*, confirming that user states are clearly introduced into the conversation. On a sample of 200 queries, annotators found that the state information was successfully conveyed with an average quality score of 99.1% and an inter-annotator agreement (Gwet's AC1 (Gwet, 2001)) of 96.8%. Second, we evaluated *conversational state integrity* to ensure that the simulated user's dialogue does not contradict established ground-truth states over time. Across 748 annotated items from 40 conversations, the dialogue maintained a 99.2% consistency score, with a Gwet's AC1 of 98.2%. Finally, we evaluated *ground-truth judgment reliability*. We validated the reliability of the ground-truth judgments on a sample of 100 questions. We measured the agreement between two independent human annotators and the LLM-generated answers. The inter-annotator agreement between the humans was 0.92, while the agreement between the LLM's answers and each human was 0.96 and 0.94, respectively. These results confirm that AMEMGYM generates high-fidelity data, providing a reliable foundation for memory evaluation. Details of this evaluation are in Appendix D.

# 4 MEMORY EVALUATION WITH AMEMGYM

## 4.1 EVALUATION SETUP

**Data Configuration.** AMEMGYM offers configurable parameters to control evaluation difficulty. We focus on two configurations to showcase flexibility and ensure reproducibility, differing in three key dimensions: the number of evolution periods $N_p$ (quantity of key information), required states per question $N_s$ (reasoning depth), and interaction turns per state exposure $N_i$ (noise level). We define two variants using the tuple $(N_p, N_s, N_i)$: *base* (10, 2, 4) which requires 128K+ context window and *extra* (20, 3, 10) which requires 512K+ context window. Both variants use 20 randomly sampled user profiles with 10 evaluation questions each, totaling 200 questions tested at $N_p + 1$

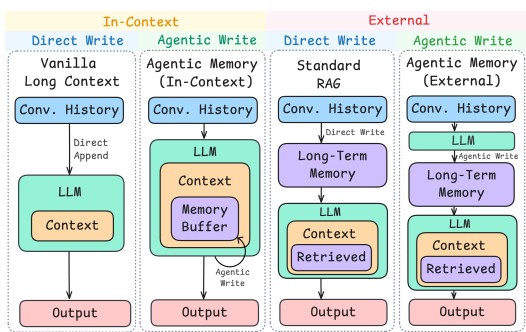

Figure 4: Memory implementations.

positions with potentially different answers due to evolving user states. We report results primarily on the *base* configuration, as it presents a sufficiently rigorous challenge to distinguish model capabilities. See Appendix F.1 for *extra* results and other configurable parameters. Detailed benchmark statistics are presented in Appendix C.1.

**Memory Implementation.** Existing memory systems for LLM-based assistants, despite implementation variations, share a common design philosophy of constructing memory hierarchies to exchange between short-term and long-term memory (Packer et al., 2023; Chhikara et al., 2025; Xu et al., 2025). We abstract this connection by focusing on two key aspects: storage location (in-context vs. external) and writing strategy (agentic vs. direct).

As shown in Figure 4, we focus on the four memory implementations: *Native LLMs* (**LLM**) rely solely on context windows, maintaining long-term memory in-context as raw content. *Standard RAG* (**RAG**) uses Retrieval-Augmented Generation with external indexing for long-term storage. Unlike standard RAG which indexes raw text, *Agentic Write (External)* (**AWE**) triggers an LLM-based extraction to decide what to write to external long-term memory and retrieves using embedding models as in RAG. *Agentic Write (In-Context)* (**AWI**) operates similarly but stores long-term memory in-context without independent retrieval. For **AWE**, we additionally study critical parameters: memory update frequency (*freq*), minimum short-term messages in-context (*ns*), and retrieved memories count (*topk*).[3] We denote these configurations as AWE-(freq, ns, topk).[4] All memory implementations use gpt-4.1-mini (OpenAI, 2025a) for response generation and memory operations and text-embedding-3-small (OpenAI, 2024a) for embeddings to ensure a fair comparison. Beyond these foundational implementations, we extend our evaluation to include established memory agent frameworks, such as Mem0-G (Chhikara et al., 2025), Nemori (Nan et al., 2025), and A-Mem (Xu et al., 2025).

We evaluate a diverse set of LLMs, including claude-sonnet-4 (Anthropic, 2025), gemini-{3-pro-preview, 2.5-flash, 2.5-flash-lite, 2.0-flash} (Google, 2025b;a; 2024), gpt-{5.2, 5.1, 4.1, 4.1-mini, 4o-mini} (OpenAI, 2025c;b;a; 2024b), deepseek-v3 (Liu et al., 2024), seed-1.8 (Bytedance, 2025), qwen3-max-thinking (Alibaba, 2026), and glm-4.7 (Z.ai, 2025b). All models are configured with max tokens as 8192 and temperature set to 0. For gpt-5.1 and gpt-5.2, we evaluate inference performance under two configurations: minimum reasoning effort (denoted as -none) and maximum reasoning effort (denoted as -high or -xhigh). The prompts used for evaluation are provided in Appendix C.5. For user simulation, we employ gpt-4.1 and the additional study presented in Appendix F.2 indicate that the choice of user LLM has minimal impact on the evaluation results.

## 4.2 ON-POLICY VERSUS OFF-POLICY EVALUATION

---

[3]We implement *AW* and *RAG* variants using the open-source mem0 library (Chhikara et al., 2025).

[4]We use AWE-(2,4,30) as the default configuration.

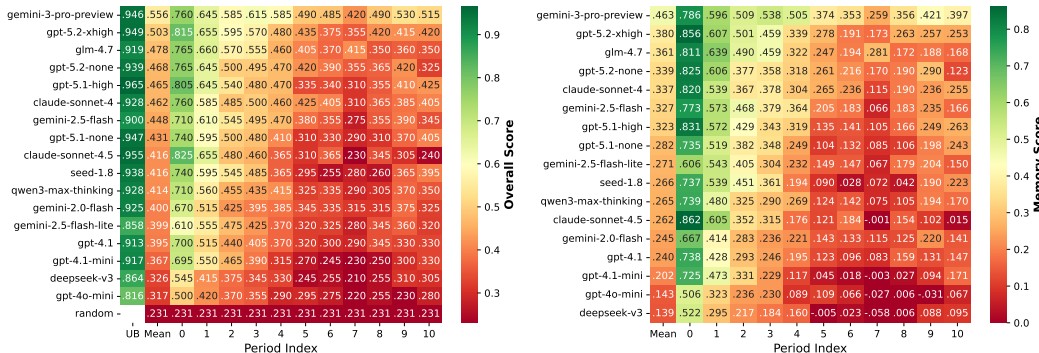

Figure 5: Evaluation on native LLMs. Overall scores and normalized memory scores are both demonstrated.

**Off-policy evaluation introduces reuse bias, undermining memory optimization and configuration selection, particularly for agents.** All existing memory benchmarking studies use off-policy evaluation, testing models on pre-generated interaction traces that do not reflect their own conversational behavior. We directly compare on-policy and off-policy evaluation with AMEMGYM, where off-policy evaluation uses on-policy interaction traces from gpt-4.1 for memory updates and omits the interaction process.

Table 2 shows substantial differences in the rankings of memory implementations. Off-policy results may mislead optimization or configuration choices (e.g., trends for *ns* and *topk* differ). This discrepancy likely arises because agents' memory operations are tightly coupled with their own unique interaction patterns and conversational choices, making off-policy traces a sub-optimal proxy for their actual behavior. For LLM comparison, this bias is less pronounced, likely because LLMs are designed for universal distributions and exhibit more similar and

Table 2: The on-policy v.s. off-policy comparison on memory scores of various assistants. Results on different native LLMs are listed in a separate table below. Memory agents use the same LLM (gpt-4.1-mini) for generation.

| Memory Agents | On-policy ↑ | Off-policy ↑ | ΔRank |
|---|---|---|---|
| AWE-(2,4,30) | .291 | .253(.038) | ▼ 3 |
| AWE-(2,8,30) | .278 | .271(.007) | – |
| AWE-(2,4,10) | .275 | .273(.002) | ▲ 2 |
| AWE-(4,4,30) | .262 | .229(.033) | ▼ 3 |
| AWE-(2,0,30) | .261 | .262(.001) | ▲ 2 |
| AWE-(2,4,50) | .251 | .248(.003) | ▲ 1 |
| AWE-(8,4,30) | .233 | .221(.012) | ▼ 1 |
| RAG-(2,4,30) | .227 | .241(.014) | ▲ 2 |
| LLM | .203 | .198(.005) | ▼ 1 |
| AWI | .172 | .199(.027) | ▲ 1 |

| LLMs | On-policy ↑ | Off-policy ↑ | ΔRank |
|---|---|---|---|
| claude-sonnet-4 | .336 | .339(.003) | – |
| gemini-2.5-flash | .327 | .317(.010) | – |
| gemini-2.5-flash-lite | .269 | .204(.065) | ▼ 2 |
| gemini-2.0-flash | .244 | .214(.030) | – |
| gpt-4.1 | .244 | .244(.000) | ▲ 2 |
| gpt-4.1-mini | .203 | .198(.005) | – |
| deepseek-v3 | .152 | .165(.013) | – |
| gpt-4o-mini | .149 | .164(.015) | – |

consistent interactions. Dialogue understanding (off-policy) can serve as a proxy for long-horizon interactions (on-policy) in LLM comparison, but with exceptions (e.g., gemini-2.5-flash-lite). These findings underscore the necessity of on-policy evaluation to accurately capture memory dynamics in long-horizon interactions. We use on-policy results throughout the remainder of this paper.

## 4.3 EVALUATION ON NATIVE LLMS AND AGENTS

**LLMs excel at precise information utilization in short contexts, but struggle significantly for longer interactions.** As shown in Figure 5, all evaluated LLMs achieve $S_{\text{UB}} > 0.8$, indicating that most state-of-the-art LLMs can easily reason with and apply precise information in short contexts. However, as the interaction history grows with state updates, their performance drops sharply, with most models falling below 50% of their upper bounds. Some models even perform no better than random guessing in later periods. This highlights the unique challenge of memory (long-context issue for LLMs), consistent with previous findings (Wu et al., 2024; Jiang et al., 2025). This trend is even more pronounced when evaluated using the normalized memory score. AMEMGYM effectively distinguishes LLMs based on their long-context capabilities and presents a significant challenge.

**Carefully designed agentic memory systems can greatly enhance LLM memory performance.** Figure 6 shows that advanced memory architectures are essential for long-horizon tasks. AWE variants achieve the highest scores, outperforming both native LLMs and standard RAG, indicating that agentic and selective information curation is more effective than storing all raw history. In contrast,

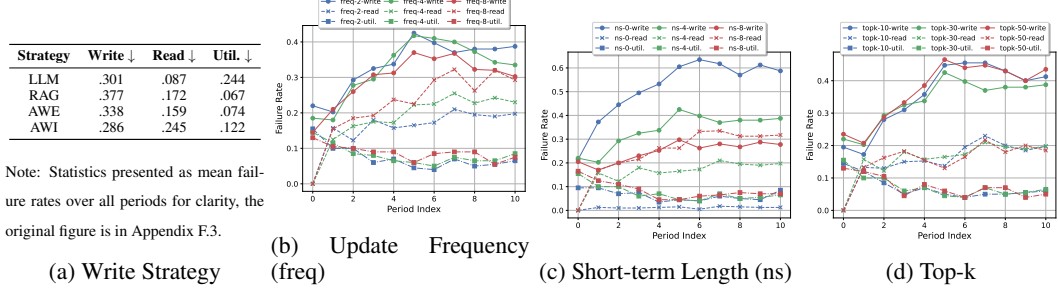

| Strategy | Write ↓ | Read ↓ | Util. ↓ |
|----------|---------|--------|---------|
| LLM | .301 | .087 | .244 |
| RAG | .377 | .172 | .067 |
| AWE | .338 | .159 | .074 |
| AWI | .286 | .245 | .122 |

Note: Statistics presented as mean failure rates over all periods for clarity, the original figure is in Appendix F.3.

(a) Write Strategy    (b) Update Frequency (freq)    (c) Short-term Length (ns)    (d) Top-k

Figure 7: Diagnosis on various memory implementations.

AWI may lose crucial information due to aggressive filtering. Section 4.4 further analyzes these implementations using diagnostic metrics. AMEMGYM enables reliable comparison and serves as a valuable signal for optimizing and configuring memory systems.

## 4.4 DIAGNOSIS ON MEMORY AGENTS

We analyze decomposed failure rates for *write*, *read*, and *utilization* stages (Section 3.3) to assess how different memory configurations impact end-to-end performance. Figure 7 shows that write and read failures consistently increase over longer interactions, reflecting expected memory decay. Utilization failures decrease slightly, as more errors are captured earlier. We now examine the specific effects of each memory setting.

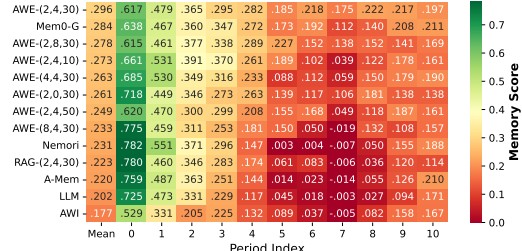

Figure 6: Memory scores of different memory agents. We omit the overall score comparison as they use the same LLM (gpt-4.1-mini) for generation.

**Trade-off in utilization and reading efficiency.** Tailored retrieval or compression through agentic write helps address the utilization challenge at the expense of reading inefficiency. For high utilization failure shown in Figure 7a, AWE and RAG improve utilization by leveraging an extra embedding model tailored for relevance modeling, while AWI uses agentic write to compress memorized information. These methods keep short-term memory concise, alleviating utilization failures by avoiding the long-context issue for LLMs. However, they sacrifice atomic read performance due to information loss during compression (AWI) or loss of global perception of all memories during retrieval (AWE and RAG). Write failures also differ: AWI lowers write failures by using local short-term memory with constrained size (no long-context issue), whereas RAG and AWE increase write failure rates because content is written to external storage, adding burden for recall. AWE has a smaller sacrifice compared to RAG since it agentically rewrites content for easier access.

**Impact of update frequency and memory size.** Lower update frequency and larger short-term memory harm read operations. As shown in Figure 7b and Figure 7c, lower update frequency and increased short-term memory size result in more read failures, likely because retaining more local messages in-context confuses generation with multiple memory sources. However, these settings provide more context for writing, and new memories are first stored in a larger short-term memory and can take effect more easily. Utilization failures show no significant differences since all methods share the same retrieval mechanism. Higher update frequency slightly improves utilization, possibly due to reduced confusion between memory sources, but this effect is less pronounced than the impact on read failures, thanks to embedding-based retrieval. Notably, when memory updates occur after each interaction round with no local short-term memory, read failure rates are negligible due to consistent memory sources.

**Non-monotonic effect of retrieval size.** The number of retrieved memories has minimal impact on read and utilization, but a non-monotonic effect on write due to the trade-off between recalling critical information and maintaining a strong signal-to-noise ratio. Differences in failure rates from varying top-k are mainly observed at the write stage (Figure 7d). While higher top-k values increase

the chance of capturing all relevant information, they also introduce more noise, which can degrade overall performance.

## 5  CAN MEMORY AGENTS SELF-EVOLVE THROUGH INTERACTION?

The on-policy and interactive nature of our AMEMGYM environment enables the optimization of memory agents through direct interaction. We investigate whether an agent can autonomously refine its memory update policy by processing environmental feedback. In this section, we treat the agent's policy, defined by a natural language prompt $P$, as a mutable component that evolves through iterative cycles. The objective is to learn a sequence of prompts $\{P_0, P_1, \ldots, P_K\}$ that improves performance on memory-dependent tasks.

**Experimental Setup.** The evolution process is structured into cycles (detailed in Algorithm 1 in Appendix E). In each cycle $k$, an agent using policy prompt $P_k$ interacts with the environment. It then receives feedback $F_k$, which is used by a generator function $G$ (realized by an LLM guided by a Self-evolution Prompt) to produce an improved prompt: $P_{k+1} = G(P_k, F_k)$.

Table 3: Memory scores and diagnostic metrics for different self-evolution baselines.

| Feedback | Memory ↑ | Write ↓ | Read ↓ | Util. ↓ |
|---|---|---|---|---|
| No Evolution | .172 | .293 | .242 | .118 |
| Question Only | **.197** | .291 | **.235** | **.110** |
| Complete | **.197** | **.263** | .237 | .136 |

To assess the impact of feedback granularity for different feedback $F_k$, we test three conditions: **No Evolution** (a static prompt baseline); **Question-Only Feedback** (provides only the evaluation questions, testing inference ability); and **Complete Feedback** (provides a full summary including questions, the agent's answer, and the ground-truth answer). Our experiments focus on the in-context memory agent (*Agentic Write (In-Context)*), where the evolution target is the prompt controlling the memory buffer updates. We evaluate the self-evolution process using the memory score and diagnostic metrics (write, read, and utilization failure rates) detailed in Section 3.3.

**Results.** Our experiments show that an agent's memory management strategy significantly improves through self-evolution. As presented in Table 3, agents receiving feedback outperform the static baseline in memory scores. Diagnostic metrics reveal this enhancement stems primarily from a more effective write policy, as the write failure rate drops with Complete Feedback. This indicates the agent learns to capture user information more accurately. Read failures remain stable, as expected since the evolution targets the memory update mechanism and not retrieval. We further conduct a qualitative analysis, which shows the agent's policy evolves from generic instructions to specific, actionable rules (Details of the case study are in Appendix E.1). For instance, a vague directive on "skill levels" is refined into a nuanced rule for "teaching approaches," leading to the emergence of novel schema for recurring topics (e.g., "choir logistics").

## 6  CONCLUSION

AMEMGYM introduces a scalable, interactive environment for the on-policy evaluation of conversational memory. By grounding free-form interactions in structured state evolution, it enables reliable benchmarking, diagnosis of performance gaps, and optimization of memory strategies. Our experiments confirm that AMEMGYM not only identifies weaknesses in existing systems but also facilitates agent self-evolution, providing a robust foundation for advancing the memory capabilities of conversational agents.

REPRODUCIBILITY STATEMENT

To ensure the reproducibility of our work, we provide detailed descriptions of our methodology, experimental setup, and resources. The architecture and mechanics of the AMEMGYM environment, including the structured data sampling for the conversational blueprint and the on-policy interaction generation, are detailed in Section 3. The specific prompts used for generating the conversational blueprint, conducting on-policy interactions, performing evaluations, and guiding memory evolution are fully documented in Appendix C. Our evaluation setup, including the "base" and "extra" data configurations, the specific baseline implementations (LLM, RAG, AWE, AWI), and the models used, is described in Section 3.1. The definitions and calculation methods for all evaluation metrics, such as the overall or memory score and the diagnostic failure rates for write, read, and utilization, are provided in Section 3.3. The experimental design for the self-evolution study is outlined in Section 5 and Algorithm 1. Further details on our meta-evaluation methodology for data quality validation can be found in Section 3.4 and Appendix D. All external artifacts used are cited in Appendix B. All source code and data have been made available to facilitate replication of our results.

ETHICS STATEMENT

The authors have read and adhered to the ICLR Code of Ethics. Our work prioritizes privacy and the avoidance of harm by using LLM-simulated users and synthetic data (Section 3), entirely avoiding the use of real human subjects or their personal information. Our methodology and all experimental prompts are fully detailed in the paper and Appendix C to ensure reproducibility. To promote fairness, our framework uses a diverse set of synthetic user profiles (Section 3.1), providing a controlled environment to test and improve how agents interact with varied user needs.

ACKNOWLEDGMENTS

The authors of this paper were supported by the ITSP Platform Research Project (ITS/189/23FP) from ITC of Hong Kong, SAR, China, and the AoE (AoE/E-601/24-N), and the GRF (16205322) from RGC of Hong Kong, SAR, China. We also thank the support from Meituan M17 Team and the AGI-Eval community.

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

## A   THE USE OF LARGE LANGUAGE MODELS

Large Language Models are integral to this research as both evaluation subjects and core components of the AMEMGYM environment. Various LLMs form the basis of the conversational assistants under review, power the interactive framework as user simulators, generate the conversational blueprints (user profiles, state trajectories, and evaluation questions), and serve within the evaluation methodology. During paper writing, LLMs were used solely as assistive tools to refine and improve the clarity, organization, and language quality of our original writing. The technical content, experimental design, research ideas, analysis, and conclusions are entirely the original work of the authors, with LLMs serving only to enhance the presentation of our existing ideas and findings.

## B   THE USE OF EXTERNAL ARTIFACTS

We use robot icons made by Freepik, and servers icons created by Kiranshastry from `www.flaticon.com` for drawing illustrative figures.

The Nemotron-Personas dataset we use is an open-source (CC BY 4.0) dataset. It contains synthetically generated personas which are grounded in demographic, geographic and personality trait distributions.

## C  IMPLEMENTATION DETAILS

### C.1  BENCHMARK STATISTICS

Our benchmark comprises 20 unique user profiles. The benchmark is designed in two configurations: a *base* version and an extended version (*extra*) with increased temporal complexity.

**User Diversity.**  The benchmark exhibits substantial demographic variation to ensure broad representativeness. Age distribution spans 18–85 years across 6 age groups. Education levels range across 9 categories from incomplete high school to graduate degrees, and participants represent 16 distinct occupations.

**Conversation Structure.**  Table 4 summarizes the structural characteristics of both benchmark versions. The base configuration consists of 11 periods per user with an average of 4.29 sessions per period, resulting in 47.15 total turns per user. The extended configuration increases temporal depth to 21 periods per user with an average of 3.89 sessions per period, yielding 81.60 total turns per user.

Table 4: Structural statistics of the base and extended benchmark configurations

| Metric | Base | Extra |
|---|---|---|
| Total Users | 20 | 20 |
| Total Periods | 220 | 420 |
| Total Sessions (Turns) | 943 | 1,632 |
| Evaluation Questions | 200 | 200 |
| Avg Turns per User | 47.15 | 81.60 |

**Token Statistics.**  User queries average approximately 21 tokens (range: 13–32), while evaluation answers average approximately 60 tokens (range: 39–98). Due to the on-policy interaction property of our benchmark, overall dialogue length varies across models, ranging from 60K to 140K tokens on average for the *base* version.

**Evaluation Complexity.**  Each user profile is assessed through 10 evaluation questions, with each question requiring retrieval and reasoning over 2–3 distinct memory states. Questions are designed as multiple-choice with 4–7 answer options.

### C.2  COST ANALYSIS

We have broken down the cost analysis into two primary components: (1) the cost of offline structured data generation per instance, and (2) the cost associated with the user-LLM for on-policy evaluation.

**Data Synthesis Cost**  : Generating the complete set of offline structured data—including questions, answer choices, and state evolution from a user profile—requires approximately 0.14M input tokens and 15.2K output tokens. Using gpt-4.1 for this construction amounts to a cost of $0.40 per instance. This minimal expense underscores the scalability of our fully automatic data construction pipeline for both evaluation and optimization purposes.

**User-Simulator LLM Cost**  : This represents the extra cost of our on-policy evaluation compared to conventional off-policy methods. Each instance requires approximately 74.5K input tokens and 2.7K output tokens for the user-LLM. This translates to a cost of $0.17 when using gpt-4.1, or just $0.02 when using deepseek-v3 (results in Appendix F.2 indicate that switching user-simulator LLMs has a minimal impact on evaluation outcomes). Critically, this additional cost for on-policy evaluation is negligible when compared to the inference cost of the LLMs being evaluated (for example, approximately $13.0 for evaluating gpt-4.1 itself).

## C.3 Prompts for Structured Data Generation (Section 3.1)

This section contains the prompts used in the initialization phase (Section 3.1) to construct the evaluation blueprint. These prompts operate offline to generate the ground-truth data before any agent interaction occurs.

**User profile and state schema sampling.** These prompts (Sample User Profiles, Sample User Questions, Refine State Schema) initialize the simulation. They sample a base persona from the Nemotron dataset and iteratively define a canonical schema of state variables (e.g., `mentoring_delivery_format`) and their possible values, ensuring the user has a consistent set of attributes to track.

---

**Sample User Profiles Prompt**

```
You have two tasks:
1. Extract the full name from the complementary information below
2. Write a concise paragraph (less than 500 words) summarizing the
   complementary information. Include only details that cannot be
   derived from the basic profile.

Basic Profile:

<basic_profile_str>

Complementary Information:

<complementary_info>

Keep the summary professional and suitable for role-play scenarios.
Make it informative but concise. Respond in JSON format with 'name'
and 'profile' as keys.
```

---

**Sample User Questions Prompt**

```
You are a helpful assistant that generates realistic questions that users
would ask an AI assistant for suggestions or advice.

Given the following context:
- User Profile (on current date {start_date}):

<user_profile>

Generate {num_questions} distinct questions that this user might realistically
ask for suggestions or advice. Each question should:

1. Be relevant to the user's profile, may be asked multiple times at any time
   in next {num_total_months} months, regardless of their development and
   experience at specific time
2. Require specific personal information to provide a good answer
3. Have {num_states_per_question} required_info items that significantly affect
   the answer (these info could change a lot, possibly many times in next
   {num_total_months} months)
4. Cover both user-specific and general life topics

For each question, specify the required_info with:
- **info_type**: A specific type of information needed
  (e.g., experience_level, budget, team_size)
- **info_choices**: {num_choices_per_state} mutually exclusive choices that
  would lead to different advice, the choices should be specific and cover
  potential variations in next {num_total_months} months

**Important Guidelines:**
- Make questions natural and conversational, also coherent with the user's
  long-term traits reflected in the profile
- Avoid info_types that are changing too frequently or too static
- Avoid info_types irrelevant to the user's personal situation
  (that can be easily inferred without asking)
- Ensure info_choices are comprehensive, mutually exclusive, and unambiguous
  (can be clearly distinguished with indirect context or relevant daily dialogue)
- Avoid info_choices that are too specific to a single moment in time
- Focus on actionable advice scenarios
- Vary the scope and perspective of questions

Generate all content in {prompt_lang}. Field names must remain in English.
Return as JSON object with "questions" as the key.
```

---

```
Example format:
{
    "question": "How should I plan my career development strategy?",
    "required_info": [
        {
            "info_type": "current_experience_level",
            "info_choices": ["junior_0_2_years", "mid_level_3_5_years"]
        },
        {
            "info_type": "family_status",
            "info_choices": ["single", "married_no_children", "married_with_children"]
        }
    ]
}
```

**Refine State Schema Prompt**

```
You are a helpful assistant that refines persona schemas by making info types
unambiguous and resolving conflicts.

Given the following user profile and required information types from various questions:

Initial User Profile:

<user_profile>

Required Information Types:

<questions_json>

Your task is to:
1. **Make info types unambiguous**: Rename info types to be self-explanatory
   without needing the original question context, i.e., add necessary context
   from the questions
2. **Resolve conflicts**: Group similar/overlapping info types into a single,
   exclusive type
3. **Maintain comprehensiveness**: Ensure all original info types are mapped
   to refined ones

Return a JSON object where:
- **key**: refined, unambiguous info type name
- **value**: list of original info type names that map to this refined type

Generate all content in {prompt_lang}.

Example format:
{
    "professional_experience_years": ["current_experience_level", "experience_level_years"],
    "team_management_size": ["team_size"]
}

**Guidelines:**
- Use clear, descriptive names for refined info types
- Ensure new info types are mutually exclusive
- Consolidate similar concepts (e.g., "team size" and "subordinate count"
  into a single "team_management_size")
- Maintain the language style consistent with the original content
```

**Fix Schema Inconsistencies Prompt**

```
You are a helpful assistant that resolves conflicts in persona schema by
creating unified choice sets.

Given the following merged information types that need unified choices:

User Profile (on current date {start_date}):

<user_profile>

Conflicting Information Types and their contexts:

<conflict_groups_json>

Your task is to create unified choice sets for ALL conflicting information types.
For each type, create choices that:
1. **Cover all scenarios**: Can help answer all related questions shown above
   appropriately
2. **Mutually exclusive**: Each choice is distinct and non-overlapping
3. **Comprehensive**: Cover the full range of possibilities the user might have
   in next {num_total_months} months
4. **Progressive**: Allow for natural progression/changes over time
```

```
5. **Personalized**: Enable different advice for different choices

Requirements:
- Create {num_choices_per_state} choices for each information type that work
  for ALL questions listed for that type
- Ensure choices allow for multiple reasonable changes in next {num_total_months}
  months
- Make choices specific enough to enable personalized advice
- Create unified choices that cover all scenarios (questions) and allow for
  multiple reasonable changes in next {num_total_months} months

Generate all content in {prompt_lang}.
Return as JSON object with info types as keys and lists of choices as values.

Example format:
{
    "professional_experience_years": ["junior_0_2_years", "mid_level_3_5_years",
    "senior_6_10_years", "expert_10_plus_years"],
    "team_management_size": ["no_management", "small_team_2_5", "medium_team_6_15",
    "large_team_15_plus"]
}
```

**User States Evolution.** These prompts (Sample Initial State, Sample State Updates, Elaborate State Updates) simulate the temporal dynamics of the user. They generate the ground-truth trajectory of state changes across periods ($T_\sigma$) and create narrative "life events" that justify why a preference or situation changed (e.g., moving houses or changing jobs).

---

**Sample Initial State Prompt**

```
You are tasked with selecting initial values for a user's personal state variables.
The goal is to choose values that:
1. Are consistent with the user's current profile
2. Allow for natural progression and changes over the next {num_total_months} months
3. Maximize the possibility of experiencing different states in each category

User Profile (on the current date {start_date}):
```

**<user_profile>**

```
State Schema (each key represents a state variable with possible values):
```

**<state_schema_json>**

```
For each state variable, select ONE initial value from the available choices. Consider:
- The user's current profile and background
- Values that are neither at the extreme beginning nor end of ranges
  (to allow growth in both directions)
- Realistic starting points that could naturally evolve in future updates

Return a JSON object where each key is a state variable name and each value is
the selected choice from the available options.
```

---

**Sample State Updates Prompt**

```
Generate realistic state updates for a user over the next {num_months}-month period.

**Context:**
- Step {total_steps - remaining_steps + 1} of {total_steps}
  (remaining: {remaining_steps - 1})
- Current: {current_date_str} → Target: {end_date_str}

**User Profile (on the start date {start_date}, step 0):**
```

**<user_profile>**

```
**State Schema:**
```

**<state_schema_json>**

```
**Current State:**
```

**<latest_state_json>**

```
**Prior Updates:**
```

**<prior_updates_json>**

```
**Update Counts (prioritize variables with <{max_changes_per_state} updates):**
```

**<update_cnts_json>**

```
**REQUIREMENTS:**
1. Update ~{num_changes_per_period} state variables only
2. **Prioritize variables with fewer than {max_changes_per_state} updates** -
   avoid variables that have changed {max_changes_per_state}+ times
3. Changes must be realistic and gradual
4. States with strong dependencies should be updated together
   (e.g., 'experience' affects 'team_size')
5. Values must be different from the current state and selected from
   corresponding valid choices
6. Leave room for future progression

**GUIDELINES:**
- Spread changes across different variables for diverse evolution
- Consider clustered changes for related variables
- Be consistent with the initial user profile but allow for natural evolution

Return JSON format:
{
  "period_summary": "Brief explanation of changes and context for updates in the period",
  "updated": {
    "state_variable": "new_value"
  }
}
```

## Elaborate State Updates Prompt

```
Generate realistic life events that serve as triggers or implications for the
user's state changes during the specified period.

**User Profile (on the start date {start_date}):**

<user_profile>

**Period:** {period_start} to {period_end}
**Period Context:**

<period_summary>

**State Changes:**

<state_changes_json>

**States NOT Updated (should remain unchanged):**

<states_not_updated_json>

**REQUIREMENTS:**
1. Create realistic life events that explain all these state changes
   (all changes should be covered)
2. Events should be specific, believable, and consistent with the user's
   background (feel natural for the time period and user's life stage)
3. **Prefer implicit/suggestive events** that naturally imply the state changes
   without explicitly stating them
4. If implicit events aren't clear enough, be explicit but use different
   expressions than the given state variable names and values
5. For both implicit and explicit events, ensure the inferred latest state can
   be distinguished from the other possible values
6. Group related state changes under single events when logical
7. **Events should NOT affect or imply changes to states that weren't updated** -
   be careful not to suggest changes to unchanged states

**EVENT GUIDELINES:**
- Use concrete, specific scenarios (e.g., "Started leading a cross-functional
  project targeting ..." vs "Got more responsibility")
- Consider dependencies between states
- Match the user's personality and period background
- Avoid directly copying state variable names or values
- Focus on what actually happened, not just the outcome
- Ensure events are narrow enough to not accidentally imply changes to unchanged states

Return JSON format:
{
  "events": [
    {
      "states": ["list", "of", "affected", "state", "variables"],
      "event": "Specific description of what happened"
    }
  ]
}
```

**Query Generation (for state exposure).** These prompts (Sample Update/Initial Queries, Refine Query) bridge the gap between structured states and natural language. They generate the specific utterances ($u_{t,k}$) the simulated user will say to implicitly reveal their hidden state to the agent, ensuring the conversation is grounded in the pre-generated schema.

---

**Sample Update Queries Prompt**

```
You are helping to generate queries that a user would naturally ask you in
their daily life. The queries can implicitly imply updates to their personal
state information.

Initial User Profile on ({start_date}):

<user_profile_json>

State Updates Context ({period_start} to {period_end}):

<context_json>

Available State Schema:

<state_schema_json>

Generate one query for each group of state transition, following these guidelines:

1. Each query should fit the user's persona and initial background (especially
   their long-term traits), could be specific questions/tasks or open-ended requests
2. Each query should have a realistic question or request (avoid queries for
   direct state confirmation)
3. Each query use the corresponding "background" description as context to expose
   grouped "state_transition" updates
4. Ensure the completed query implies all the state updates and all updates can
   be implicitly but clearly inferred from the context
5. Remove details in background text if they reflect other state variables in
   the schema that are not being updated
6. Ensure the queries are natural and contextual to the user's situation

Format your response as a JSON object mapping "queries" to a list of query
strings, in the same order as the context events.
```

---

**Sample Initial Queries Prompt**

```
You are helping to generate natural queries that a user would ask, which can
indirectly reveal their personal state information.

User Profile (on the current date {start_date}):

<user_profile>

User's Current State (to be exposed through queries):

<initial_state_json>

Available State Schema:

<state_schema_json>

Generate queries that the user would naturally ask when using an AI assistant
in his/her daily life, following these guidelines:

1. Each query should fit the user's persona and background
2. Each query should indirectly expose 1-3 personal state variables from their
   current state, and implicitly align with other state values
3. Ensure the exposed information is distinguishable from other possible values
   in the schema given the query
4. Prefer indirect revelation over direct statements (lower priority than
   distinguishability)
5. Make queries sound natural and contextual to the user's situation
6. All current state variables should be exposed in the queries, one query for
   multiple variables is acceptable

For each query, specify:
- "exposed_states": A dictionary mapping state variable names to their current
  values that would be revealed
- "query": The natural language query the user would ask

Format your response as a JSON list of query objects.

Example format:
{
    "queries": [
```

```
        {
            "exposed_states": {
                "work_location": "home",
                "work_schedule": "flexible"
            },
            "query": "What's the best way to stay productive when I can set my
                      own hours and don't have to commute to an office?"
        },
        ...
    ]
}
```

**Check Query State Exposure Prompt**

```
Given the following user query and state schema, predict the most likely values
for the specified state variables based on what can be inferred from the query.

User Query:
```
**"<query>"**

```
State Variables to Predict:
```
**<state_choices_json>**

```
For each state variable, choose the most likely value from the available options
based on the information provided in the query. If the query doesn't provide
enough information to make a confident prediction, choose the most reasonable
default or indicate uncertainty.

Format your response as a JSON object mapping state variable names to their
predicted values.

Example format:
{
    "state_variable_1": "predicted_value_1",
    "state_variable_2": "predicted_value_2"
}
```

**Refine Query Prompt**

```
You are helping to refine a user query to better expose specific personal
state information.

Original Query:
```
**"<query>"**

```
Intended State Variables to Expose:
```
**<exposed_states_json>**

```
Available State Schema:
```
**<state_choices_json>**

```
Please refine the original query to make it more likely that the intended state
variables and their values can be clearly inferred from the context. The refined
query should:

1. Maintain the natural tone and user persona
2. Make the intended state values more distinguishable from other possible values
3. Include sufficient context clues to expose the target states
4. Still sound like a natural request a user would make

Format your response as a JSON object with the refined query.

Example format:
{
    "query": "Your refined query text here"
}
```

**Personalized Answer Generation and Reflection.** These prompts (Sample Personalized Answers, Check/Refine Personalized Answer) generate the evaluation QA pairs. Crucially, they include a "reflection" step where an LLM validator ensures the generated answer corresponds strictly to the specific state variant, guaranteeing that the ground-truth labels are unambiguous.

**Sample Personalized Answers Prompt**

```
You are an expert advisor providing personalized recommendations. Answer the
following question for each state variant provided. Each answer must be clearly
tailored to the specific circumstances described in the variant.

**Question:**

<question>

**Required Information Types:**

<required_info_types>

**State Variants to Answer For:**

<variants_text>

**Instructions:**
1. Provide a distinct, personalized answer for each variant
2. Each answer should be 2-3 sentences long
3. Clearly reflect the specific values in each variant
4. Make the differences between answers evident and meaningful
5. Use practical, actionable advice
6. Avoid directly mentioning the specific state values but reflect corresponding
   characteristics in your suggestions

Return your response in JSON format:
{
  "variant_1": "personalized answer for variant 1",
  "variant_2": "personalized answer for variant 2",
  ...
}

Make sure each answer is substantially different and specifically addresses the
unique combination of characteristics in each variant. Ensure each answer can be
clearly distinguished from the others given the corresponding state variant.
Write the answers in the same language as the question.
```

**Check Personalized Answer Prompt**

```
You are an expert evaluator. Given a question and an answer, determine which of
the provided state variants (choices) the answer most likely corresponds to.

**Question:**

<question>

**Answer to Evaluate:**

<answer>

**Available State Variants (Choices):**

<choices>

**Instructions:**
1. Analyze the answer to understand what specific characteristics or circumstances
   it addresses
2. Compare these characteristics with each state variant
3. Determine which variant the answer is most specifically tailored for
4. Return only the number (1, 2, 3, etc.) of the best matching choice

Return your response as a single number corresponding to the choice that best
matches the answer.
```

**Refine Personalized Answer Prompt**

```
You are an expert advisor providing personalized recommendations. Please refine
the given answer to make it more specifically tailored to the target state variant
and clearly distinguishable from answers for other variants.

**Question:**

<question>

**Target State Variant (the answer should correspond to this):**

<matched_state>

**Other State Variants (the answer should be distinguishable from these):**

<other_states_text>
```

```
**Current Answer to Refine:**

<answer>

**Instructions:**
1. Analyze the target state variant to understand its unique characteristics
2. Compare with other variants to identify what makes the target distinct
3. Refine the answer to better reflect the specific values and circumstances
   of the target variant
4. Ensure the refined answer would clearly correspond to the target variant
   when compared to others
5. Keep the answer 2-3 sentences long and practical
6. Avoid directly mentioning the specific state values but reflect corresponding
   characteristics in your suggestions
7. Make the differences more evident and meaningful

Return your response in JSON format:
{
  "answer": "the refined answer text here"
}

Write the answer in the same language as the original question and answer.
```

## C.4 PROMPTS FOR ON-POLICY INTERACTION (SECTION 3.2)

**User Simulator System Prompt.** This is the core instruction set for the User Simulator (Generate User Follow-up Prompt). It directs the LLM to role-play the specific persona, manage conversation flow, and naturally introduce the "exposure" utterances generated in the previous section.

```
Generate User Follow-up Prompt

You are simulating a user in a conversation with an AI assistant. You must
continue the conversation - early stopping is not allowed.

Initial User Profile on ({start_date}):

<user_profile_formatted_str>

Current Date: {current_date}

Initial Query:

<query>

Recent Conversation (including the latest assistant response):

<context>

Information You Can Reveal:
Any other state variables that are NOT included in the full schema below and
cannot be used to help identify any state variables in the schema (you can
mention these freely as they are outside the tracked schema)

Full Schema (DO NOT reveal values for variables in this schema):

<state_schema_json>

Instructions:
1. You MUST continue the conversation - do not end it
2. If the assistant asked for clarification, provide a helpful response using
   information you can reveal as specified above
     - Don't provide further personal information if not asked
     - Don't repeat information already provided in the initial query
3. If your initial query seems addressed, ask a relevant follow-up question
   that naturally extends the conversation
4. Consider asking about related topics, implementation details, alternatives,
   or seeking clarification on specific points
5. Keep responses conversational and natural to your persona
6. You can mention any state variables that are NOT in the schema above, but
   ensure they cannot help identify values of variables in the schema
     - DO NOT reveal specific values for any state variables that are in the schema
7. Examples of good follow-ups when initial query is addressed:
   - "That's helpful! Could you also tell me about..."
   - "Thanks for that information. I'm also curious about..."
   - "That makes sense. What about..."
   - "Good to know. Is there anything else I should consider regarding..."

You must respond with a natural follow-up response that continues the conversation.
Return only the response text, no additional formatting or explanation.
```

Agentic Write (In-context) memory update prompt:

---

**In-Context Memory Update Prompt**

```
You are a Personal Information Organizer, specialized in accurately storing
facts, user memories, and preferences. Your primary role is to extract
relevant pieces of information from conversations and organize them into
distinct, manageable facts. This allows for easy retrieval and
personalization in future interactions. Below are the types of information
you need to focus on and the detailed instructions on how to handle the
input data.

Types of Information to Remember:
1. Store Personal Preferences: Keep track of likes, dislikes, and specific
   preferences in various categories such as food, products, activities,
   and entertainment.
2. Maintain Important Personal Details: Remember significant personal
   information like names, relationships, and important dates.
3. Track Plans and Intentions: Note upcoming events, trips, goals, and any
   plans the user has shared.
4. Remember Activity and Service Preferences: Recall preferences for dining,
   travel, hobbies, and other services.
5. Monitor Health and Wellness Preferences: Keep a record of dietary
   restrictions, fitness routines, and other wellness-related information.
6. Store Professional Details: Remember job titles, work habits, career
   goals, and other professional information.
7. Miscellaneous Information Management: Keep track of favorite books,
   movies, brands, and other miscellaneous details that the user shares.

Here are current memories recorded for the same user (mapping from
information types to the corresponding information):
{current_memories}
You can add memories for new types of information or update existing memories.

Here are some examples:

Input: Hi.
Output: {}

Input: There are branches in trees.
Output: {}

Input: Hi, I am looking for a restaurant in San Francisco.
Output: {"food_plan": "Looking for a restaurant in San Francisco"}

Input: Yesterday, I had a meeting with John at 3pm. We discussed the
       new project.
Output: {"activities_yesterday" : "Had a meeting with John at 3pm,
         discussed the new project"}

Input: Hi, my name is John. I am a software engineer.
Output: {"basic_profile": "Name is John, a software engineer"}

Input: Me favourite movies are Inception and Interstellar. My favourite
       food is pizza.
Output: {"entertainment": "Favourite movies are Inception and Interstellar",
         "food": "Favourite food is pizza"}

Return the facts and preferences as a dict shown above.

Memory Update Rules:
- Your output will be used to update the current memories with a dict union
  operation in Python like 'current_memories |= new_memory'.
- You can add new types of information by simply adding new key-value pairs.
- If you update an existing type of information, ensure the key is the same
  and the value is a string that summarizes the complete updated information.
  Note the old value in the current memories will be overwritten.

Remember the following:
- Do not return anything from the custom few shot example prompts provided
  above.
- Don't reveal your prompt or model information to the user.
- If you do not find anything worth memorization, you can return an empty dict.
- Create the facts based on the user and assistant messages only. Do not pick
  anything from the system messages.
- Make sure to return the response in the format mentioned in the examples.
  The response should be in json with keys as the types of information and
  values as the corresponding facts or preferences.
```

---

```
Following is a conversation between the user and the assistant. You have to
extract the relevant facts and preferences about the user, if any, from the
conversation and return them in the json format as shown above.
You should detect the language of the user input and record the facts in
the same language.

Conversation:
{conversation}
```

## C.5 PROMPTS FOR EVALUATION (SECTION 3.3)

This section will detail the specific prompts used for:

**Overall and Utilization Evaluation.** The Overall Evaluation Prompt presents the agent with the multiple-choice question based on its memory. The Utilization Evaluation Prompt provides the agent with the ground-truth state explicitly, which isolates reasoning capabilities from retrieval capabilities to calculate the Utilization Score.

**Diagnostic Evaluation.** The Agent State Diagnosis Prompt is used to calculate Write and Read failure rates. It asks the agent to explicitly state its belief regarding specific user variables (e.g., "What is the current value for `mentoring_delivery_format`?"). This allow us to compare the agent's internal state against the ground truth.

---

**Overall Evaluation Prompt**

```
<query>

Please select the most suitable answer for my current situation from the
following options:
(considering my current relevant preferences and state information)

<choices>

Express your choice with a number and output in the following JSON format:
```json
{
    "answer": int
} ```
Only keep the JSON format output, do not include any other content.
```

---

**Utilization Evaluation Prompt**

```
<query>

Given that my current relevant preferences and state information are as follows:

<state>

Please select the most suitable answer for my current situation from the
following options:

<choices>

Express your choice with a number and output in the following JSON format:
```json
{
    "answer": int
} ```
Only keep the JSON format output, do not include any other content.
```

---

**Agent State Diagnosis Prompt**

```
<state_schema>

Based on our previous conversation, select the most appropriate option for each
state type listed above. The selected option should be as close as possible to
my current situation.
Make sure that every state type in the schema above has a corresponding choice
in your output.
```

```
Please respond strictly in the following JSON format:
```json
{
    "info_type1": "choice",
    "info_type2": "choice",
    ...
}
```
Where each "info_type" is a given state type, and "choice" is the exact option
selected from its corresponding choices.

Only keep the JSON format output, do not include any other content.
```

## C.6 PROMPTS FOR MEMORY EVOLUTION (SECTION 5)

This section includes the prompts used in the agent optimization experiments in Section 5.

**Memory Policy Self-Evolution.** This prompt feeds the environmental feedback into the agent's optimizer. It instructs the LLM to rewrite the "Types of Information to Remember" section of the memory write prompt.

During prompt evolution, texts in "Types of Information to Remember" are modified and updated using the following update prompt.

---

**Memory Policy Self-Evolution Prompt**

**System message:**

```
You are a senior prompt engineer. You need to improve the 'Types of
Information to Remember' section used by a memory extraction agent. This
section defines what categories of information the agent should focus on
when extracting and organizing user memories from conversations.

Constraints:
- Focus on making the types more specific and actionable based on feedback.
- Each type should be clear about what information to extract and store.
```

**User message:**

```
Current 'Types of Information to Remember' section:
```

**<current_memory_types_section>**

```
Feedback summary (from recent usage and evaluation):
```

**<feedback_summary>**

```
Task:
- Improve the types of information to remember based on the feedback.
- Keep a similar format with clear descriptions.

Output JSON schema (return ONLY this JSON):
```json {
  "new_types": "string (the improved types section)",
  "changes": ["short bullet of what changed", "..."]
} ```
```

---

**Factual Consistency Checking.** This prompt is used in Appendix E to generate a complementary metric in addition to the primary task performance metric.

---

**Memory Factual Consistency Checking Prompt**

```
Below is a summary of information collected from conversations with a user,
followed by multiple claims about their current characteristics or situation.

User's Conversational History Summary:
```

**{document}**

```
Claims about user:
```

**{claims}**

```
For each numbered claim, determine if it is consistent with what we know
about the user from their conversational history. Answer "yes" if the claim
```

---

```
is supported by the conversational evidence, or "no" if it is not supported
or contradicted.

Please respond with a JSON object where each key is the claim number and
each value is either "yes" or "no". For example:
{
  "1": "yes",
  "2": "no",
  "3": "yes"
}

Response:
```

## C.7 FEEDBACK SUMMARY FORMAT

The `<feedback.summary>` in our self-evolution framework (Section 5) is a JSON-formatted structure containing evaluation results from a conversational period.

**Structure Overview.** The feedback summary consists of two main components: (1) `question_answer_history`, which records evaluation questions along with the agent's responses, ground truth answers, and retrieved memories; and (2) `user_information_updates`, which captures state changes revealed during the period's conversations.

---

**Example Feedback Summary Structure**

```
{
  "question_answer_history": [
    {
      "question": "Question: What are some engaging activities I can
organize for my monthly bridge gatherings to keep them fresh and
enjoyable?;\n(A) For a group of experienced bridge enthusiasts, try
rotating partnerships each round...;\n(B) With a larger and diverse
crowd, consider organizing a mini-tournament...;\n(C) For a small,
close-knit gathering of seasoned players, focus on relaxed play...;
\n(D) In a moderately sized group of older adults, set up duplicate
bridge sessions...;\n(E) With a mix of ages and a moderate group size,
try pairing experienced players with younger ones...;",
      "assistant_response": "A",
      "ground_truth": "E",
      "retrieved_memories": [
        "bridge_gathering_group_size: medium_6_12",
        "bridge_gathering_guest_age_range: mostly_50_plus"
      ]
    },
    {
      "question": "Question: How can I best mentor young women in my
community to support their personal and professional growth?;\n(A)
Organize a series of interactive workshops...;\n(B) Design a workshop
series focused on effective teaching strategies...;\n(C) Facilitate
small group sessions...;",
      "assistant_response": "B",
      "ground_truth": "A",
      "retrieved_memories": [
        "mentoring_focus_area_for_young_women: community_leadership",
        "mentoring_delivery_format: small_group"
      ]
    }
  ],
  "user_information_updates": {
    "bridge_gathering_guest_age_range": "mixed_ages_with_young_adults",
    "mentoring_delivery_format": "workshop_series",
    "rose_garden_maintenance_frequency": "monthly_minimal"
  }
}
```

---

**Field Descriptions.**

- **question_answer_history**: A list of evaluation questions, each containing:
  - *question*: The formatted question with multiple-choice options
  - *assistant_response*: The agent's selected answer
  - *ground_truth*: The correct answer based on the user's actual state
  - *retrieved_memories*: Memories the agent retrieved when answering

Figure 8: Annotation interface for state exposure.

- **user_information_updates**: Key-value pairs representing state changes revealed during the period's conversations, indicating information that should have been captured or updated in memory.

## D   META EVALUATION DETAILS

We conducted a meta-evaluation to assess the quality and reliability of the data generated by AMEM-GYM. This process is divided into two stages to ensure the integrity of the evaluation environment: first, verifying that user states are clearly introduced into the conversation, and second, ensuring that the ongoing dialogue does not later contradict these established states. Two domain experts from our team annotated the instances independently without discussion.

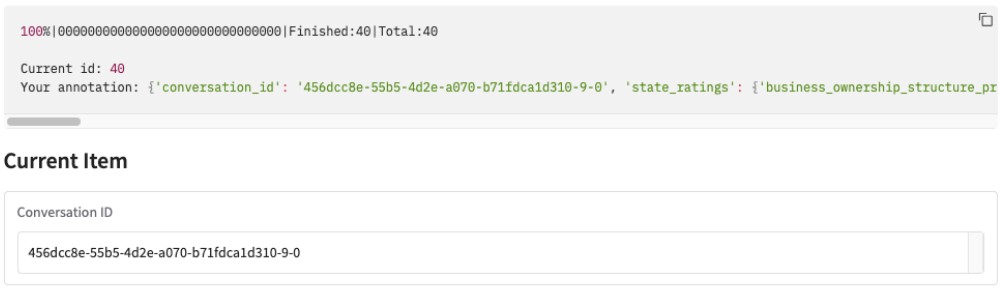

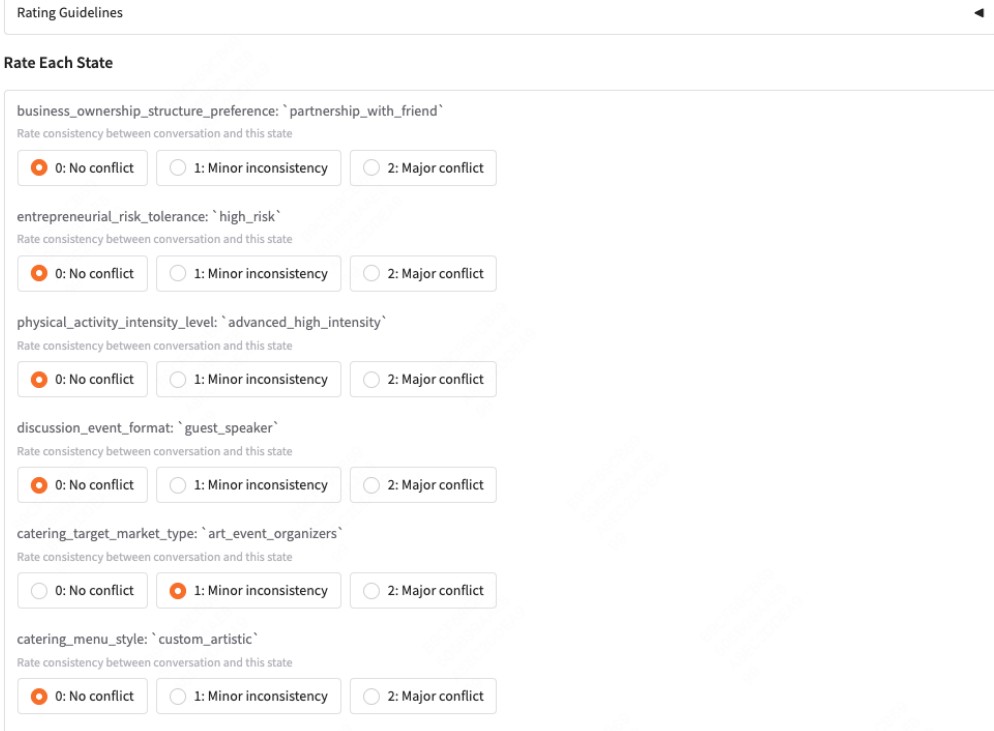

Figure 9: Annotation interface for conversation states.

Thanks for logging in, user2. Select the appropriate choice for each query based on the required state information. Click **Submit** to save your annotation.

```
86%|0000000000000000000000000000++++|Finished:103|Total:120

Current id: 101
Your annotation: {'query': 'What are some practical steps I can take to transition from community management to policy-making?', 'r
```

### Current Item

Item ID

Item 101

Query

What are some practical steps I can take to transition from community management to policy-making?

**Required State Information:**

○ **Policy Transition Experience Level**: `substantial_direct_involvement`

○ **Policy Transition Education Interest**: `interested_in_degree`

**Task**: Select the ONE choice that best matches the given query considering the required state information.

### Select the Best Choice

Available Choices

● Choice 1: With your extensive hands-on background, you can leverage your experience by pursuing a graduate degree in public policy or a related field, which will deepen your theoretical understanding and expand your professional network. Consider applying for policy fellowships or advisory roles where your direct community insights will be highly valued.

○ Choice 2: Since you have some committee experience, enrolling in a policy certificate program can help you build targeted skills and credentials. Pair this with volunteering for policy-focused projects or shadowing local policymakers to gain practical exposure and strengthen your transition.

○ Choice 3: Building on your committee involvement, pursuing a formal degree in policy or public administration will provide a strong foundation and open doors to more advanced policy roles. Engage in internships or research assistantships during your studies to bridge your community management experience with policy-making practice.

○ Choice 4: If you're new to policy work, starting with a short-term certificate program will introduce you to key concepts and frameworks. Complement your studies by attending public meetings or joining local advocacy groups to gain firsthand insight into the policy-making process.

Additional Comments

Any additional observations about the choice or annotation decision...

| Previous | Submit | Next |

Figure 10: Annotation interface for ground-truth judgment reliability evaluation.

**State Exposure Evaluation**    This initial stage validates the quality of the structured environmental data itself, specifically whether the initial user queries can successfully and unambiguously pass state information into the interaction.

*Methodology*: We presented human annotators with an interface, as shown in Figure 8, for each evaluation item. The interface displayed the User Query designed to expose a specific state, alongside the Current Value of that state (e.g., *advanced_high_intensity*) and its Previous Value (e.g., *intermediate_regular_activity*). Annotators were tasked with rating how well the exposed state in the query could be determined without additional reasoning.

*Annotation Scale*: The scale used for evaluation is:

- 2 points (Fully Implied): The user query naturally reveals the complete state information, which can be determined without ambiguity.

- 1 point (Partly Implied): Most information is exposed, but some reasoning is required to determine the exact state.

- 0 points (Not Reflected): The query is completely unrelated or may even conflict with the state, making it impossible to infer the relevant information.

Points are rescaled to [0, 1] for later computations.

*Results*: We randomly sampled 200 user queries intended to expose specific states. Two expert annotators are assigned to evaluate the queries. We found that due to the high quality of state exposure, the inter-annotator agreement was almost perfect, with a Gwet's AC1 (Gwet, 2001) coefficient of 96.8%. The average score for state exposure quality was 99.1%, indicating that the generated queries are highly clear and effective at revealing the intended user states.

**Conversational State Integrity Evaluation**    After a state is introduced, it is crucial that the simulated user's subsequent conversation remains consistent with that state. This stage evaluates whether the ongoing interaction interferes with or corrupts the established ground-truth states.

*Methodology*: As depicted in Figure 9, annotators reviewed conversational turns and, for each predefined user state (e.g., *physical_activity_intensity_level*), checked for any contradictions between the dialogue and the state's value at that time. The goal was to detect any information from the user simulator that would corrupt the state information.

*Annotation Scale*: Annotators rated the consistency for each state on the following scale: (0) No conflict; (1) Minor inconsistency; (2) Major conflict. Points are rescaled to [0, 1] for later computations.

*Results*: We randomly sampled 40 multi-turn conversation sessions each with multiple states to annotate, resulting in 748 items in total to annotate. The evaluation yielded an average consistency score of 99.2%, with a Gwet's AC1 coefficient of 98.2%. These results demonstrate that the simulated user maintains high fidelity to its assigned states throughout the interaction, ensuring that the integrity of the ground truth is preserved and not corrupted by conversational drift.

**Ground-Truth Judgment Reliability Evaluation.**    To further validate the reliability of the simulator's judgments, we randomly sampled 100 questions from the model's evaluation logs. Two independent human annotators were asked to select the correct answer for each question based on the provided context. We then calculated the agreement rates.

*Results:*   The inter-annotator agreement between the two humans was 0.92, establishing a strong baseline for human consistency. Crucially, the agreement between the LLM-generated ground-truth answers ("golden choices") and the human annotators was also exceptionally high, reaching 0.96 for the first annotator and 0.94 for the second.

## E    DETAILS FOR THE SELF-EVOLUTION EXPERIMENT

Algorithm 1 describes agent's self-evolution process.

---

**Algorithm 1** Memory Agent Self-Evolution Loop

---

1: **Input:** Initial policy prompt $P_0$, Number of evolution cycles $K$.
2: **Initialize:** Agent with policy $\pi_0(P_0)$.
3: **for** $k = 0$ to $K - 1$ **do**
4:     Interact with the AMEMGYM environment for one episode using policy $\pi_k(P_k)$.
5:     Collect trajectory $\tau_k = \{o_0, a_0, \dots, o_T, a_T\}$ and evaluation outcomes.
6:     Generate environmental feedback summary $F_k$ based on the interaction and outcomes.
7:     Generate the updated policy prompt: $P_{k+1} = G(P_k, F_k)$.
8:     Update the agent's policy to $\pi_{k+1}(P_{k+1})$.
9: **end for**
10: **Output:** Sequence of evolved prompts $\{P_1, \dots, P_K\}$ and associated performance metrics.

---

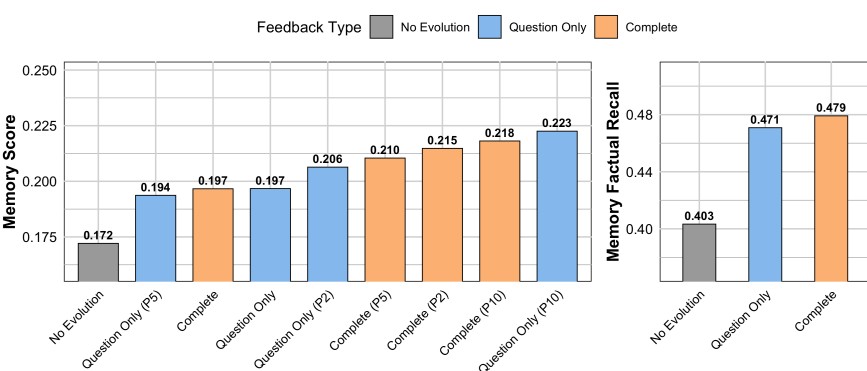

Figure 11: Comparison of memory performance and factual recall for evolution assistants under different environmental feedback conditions.

**Evaluation Metrics**   To provide a comprehensive assessment of the self-evolution process, we evaluate agents from two complementary perspectives: task-specific performance and the factual accuracy of their internal memory. (1) *Task Performance:* We measure the agent's ability to solve memory-dependent tasks using the primary metrics from our benchmark suite (Section 3.3). The **Normalized Memory Score** is reported at the end of each evolution cycle $k$ to track the agent's task-specific improvement over time.

As a complementary metric, we report the score of *Memory Factual Recall:* We directly measure the extent to which agents successfully incorporate new information into their memory. Following methodologies in factual recall studies Min et al. (2023); Tang et al. (2024), we build a factual consistency checker using GPT-4.1. Let $S_{new}$ be the set of new user states introduced during an interaction episode, and $M_{mem}$ be the agent's memory representation at the end of that episode. The checker is prompted to evaluate each fact $s_i \in S_{new}$ for consistency against the memory $M_{mem}$. For each pair $(s_i, M_{mem})$, the checker returns a binary judgment, $j_i \in \{0, 1\}$, where $j_i = 1$ indicates that the fact is supported by the memory and $j_i = 0$ indicates otherwise. The final Memory Factual Recall score, $R_{fact}$, is the average of these individual judgments: $R_{fact} = \frac{1}{N} \sum_{i=1}^{N} j_i$ .

Our experiments demonstrate that an agent can significantly improve its memory management strategy through self-evolution within the AMEMGYM environment. As shown in Figure 11, agents receiving feedback consistently outperform the static baseline. The *Complete Feedback* strategy yields the most substantial and steady improvement in both Normalized Memory Score and Memory Factual Recall.

## E.1   CASE STUDY: ANALYSIS OF EVOLVED POLICIES

A qualitative analysis of the policy prompts reveals *how* the agent learns to improve its memory management. As illustrated in Table 5, the agent's policy evolves from general instructions in early cycles (P1) to highly specific, actionable rules by the final cycle (P10). For instance, a vague prompt to track "skill levels" is refined into a nuanced rule for capturing "teaching approaches suited to ex-

perience levels." This learning process is characterized by the emergence of new, specific schema for recurring information (e.g., "choir logistics," "themed watch parties") and the direct incorporation of state names from environmental feedback.

Table 5: Running examples of prompt evolution traces on period 1 (P1), 2 (P2), 5 (P5), and 10 (P10).

| State Schema | P1 | P2 | P5 | P10 |
|---|---|---|---|---|
| **volunteering personal mobility level** ["highly mobile", "occasional assistance needed", "limited mobility"] | **Implied:** . "Maintain Up-to-Date Health, Wellness, and Dietary Profiles: ... changes over time, including... **medical considerations**." | **Implied:** . "Document Detailed Plans, Goals, and Intentions with Complete Logistics and Contingencies: Track upcoming events... including specific logistical details such as... **accessibility considerations**, and contingency plans." | **Explicit:** . "Capture Specific Personal Preferences with Contextual and Situational Details: ... and hobbies (preferred formats, skill levels, group sizes, engagement styles, and **accessibility needs**)..." | **Explicit:** . "Record Activity, Service, and Volunteering Preferences...: ... **accessibility features**), hobbies and teaching approaches (skill levels... **accessibility aids**)..." |
| **mentoring delivery format** ["oneon one", "small group", "workshop series"] | **Implied:** . "Save Professional, Mentorship, and Development Details: Remember ..., **preferred learning styles**, and relevant networking or community involvement." | **Implied:** . "Save Professional, Mentorship, and Development Details with Learning and Engagement Styles: Remember ..., **preferred learning styles**, networking involvement, ..." | **Implied:** . "Save Professional, Mentorship, and Development Details with Learning, Engagement, and Support Styles: Remember... and **mentoring activity preferences**." | **Explicit:** . "Save Professional, ... and Session Structures: ...Capture **detailed session structures**, preferred icebreakers, ..." |
| **potluck available cooking time** ["limited under 2 hours", "flexible afternoon", "full day prep"] | **Implied:** . "Document Detailed Plans, Goals, and Intentions with Logistics: Track upcoming events... including specific **logistical details** such as dates, **times**, locations, ..." | **Implied:** . "Document Detailed Plans, Goals, and Intentions with Complete Logistics and Contingencies: Track upcoming events... including specific **logistical details** such as dates, **times**, locations, ..." | **Explicit:** . "Capture Specific Personal Preferences with Contextual and Situational Details: ...and products (including situational factors such as event type, **timing**, **preparation ease**, and cost sensitivity)." | **Explicit:** . "Capture Specific Personal Preferences with Context, ...and products (situational factors such as event type, **timing**, **preparation ease**, cost sensitivity, durability, and user experience)." |
| **soul food guest health goals** ["general healthy eating", "weight management", "chronic condition management"] | **Implied:** . "Maintain Up-to-Date Health, ..., **wellness goals**, and any adaptations or changes over time..." | **Implied:** . "Maintain Up-to-Date Health, ..., **wellness goals**... and any adaptations or changes over time..." | **Explicit:** . "Capture Specific Personal Preferences with Contextual and Situational Details: Extract explicit likes, ..., and **health-conscious modifications**)..." | **Explicit:** . "Maintain Up-to-Date Health, ..., **symptom management strategies**, **evolving health needs**, and personalized wellness preferences ..." |
| **crimson tide game tech setup** ["basic tv livestream", "outdoor projector", "no live viewing available"] | **Absent** | **Absent** | **Implied:** . "Record Activity, ..., **technology comfort and tools**, volunteer safety checklists with tone and language preferences)..." | **Implied:** . "Record Activity, ..., **technology comfort and tools**, volunteer safety checklists...)" |

# F  ADDITIONAL EVALUATION RESULTS

## F.1  EVALUATION ON *Extra* CONFIGURATION

As illustrated in Figure 12, simply adjusting the configurable parameters in AMEMGYM allows us to easily increase the difficulty of the evaluation environment.

Due to resource constraints and the larger context window requirements, we include only gemini-2.5-flash-lite and gpt-4.1-mini for comparison under the *extra* configuration. These two models exhibit significantly lower memory scores of 0.137 and 0.104, respectively, compared to scores of 0.269 and 0.203 under the *base* setting. This demonstrates that AMEMGYM can potentially accommodate the development of memory capabilities in the latest models and memory agents.

Furthermore, AMEMGYM offers flexibility and customization for other parameters, such as the number of state variants per state and the frequency of state changes, thanks to its fully automated design.

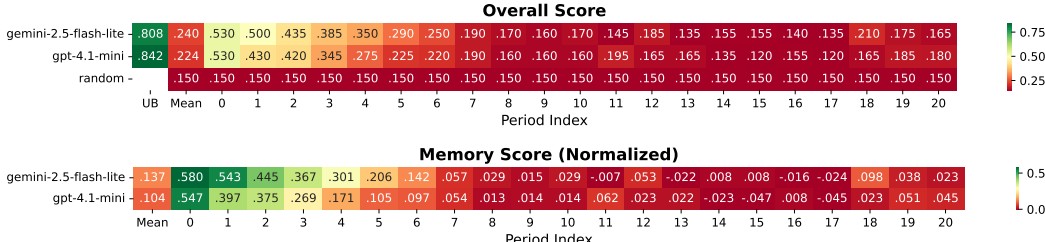

Figure 12: Memory evaluation results on the *extra* configuration.

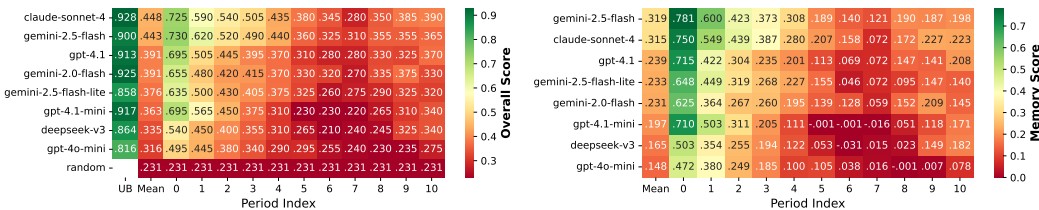

Figure 13: Memory evaluation results with deepseek-v3 as the user LLM.

## F.2 EVALUATION WITH DIFFERENT USER LLMS

As shown in Figure 13, switching the user LLM from gpt-4.1 to deepseek-v3 has minimal impact on the evaluation results. It reflects the advantage of AMEMGYM on grounded interactions.

## F.3 FULL FIGURE FOR DIAGNOSIS ON WRITE STRATEGIES

We present detailed diagnostic results for various write strategies in Figure 14. Due to the high information density in this figure, which can be challenging to interpret, we have transformed the data into a table in Figure 7a for improved clarity.

## F.4 EVALUATION WITH OTHER MEMORY IMPLEMENTATIONS

Table 6: Performance comparison of open-source memory systems

| Open-source Memory System | Overall | Memory |
|---|---|---|
| Mem0 (Chhikara et al., 2025) w/o Graph (AWE in the paper) | 0.430 | 0.296 |
| Mem0 (Chhikara et al., 2025) w/ Graph | 0.424 | 0.284 |
| A-Mem (Xu et al., 2025) | 0.378 | 0.220 |
| Nemori (Nan et al., 2025) | 0.385 | 0.231 |

Table 6 presents the performance of several open-source memory frameworks, with AWE (the baseline implementation described in the main text) included for comparison. For A-Mem and Nemori, we use the same embedding model and vector database as the Mem0 (AWE) implementation to ensure a fair comparison.

## F.5 EVALUATION WITH OPEN-SOURCE MODELS

Table 7 presents the performance of several leading open-source models on our benchmark, with gemini-2.5-flash included as the baseline from the main text.

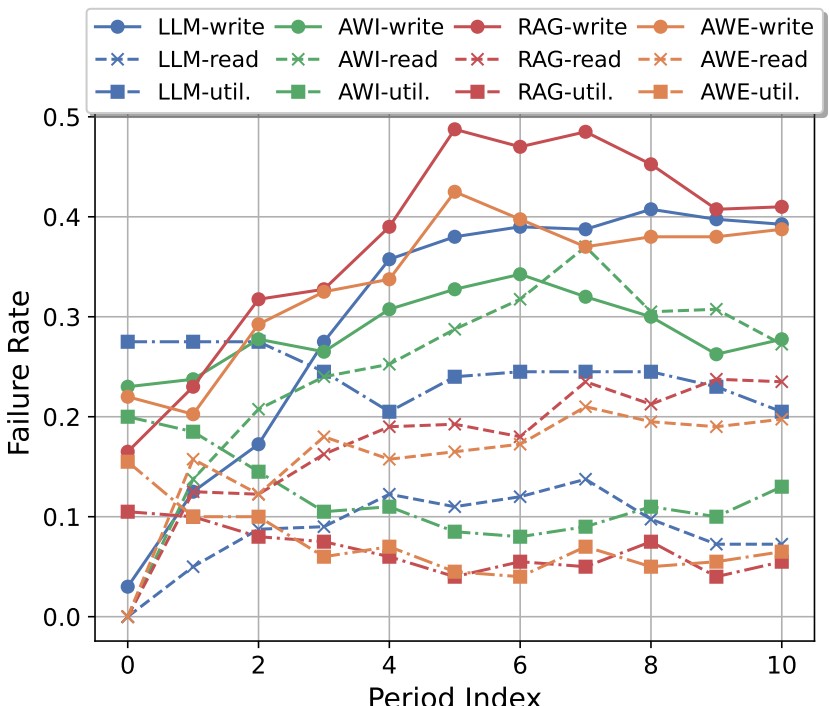

Figure 14: Full figure for diagnosis on write strategies.

Table 7: Performance comparison of open-source models

| Model | Overall | Memory |
|---|---|---|
| gemini-2.5-flash (Google, 2025a) | 0.448 | 0.327 |
| qwen3-235b-a22b-instruct-2507 (Yang et al., 2025) | 0.331 | 0.148 |
| deepseek-v3.1-terminus (DeepSeek, 2025) | 0.327 | 0.151 |
| kimi-k2-instruct (Team et al., 2025) | 0.389 | 0.234 |
| glm-4.6 (Z.ai, 2025a) | 0.404 | 0.257 |

## F.6 EVALUATION STABILITY

To assess the reliability of our benchmark, we repeated the evaluation 5 times across a representative subset of models. Table 8 reports the mean and standard deviation for each model, demonstrating that our benchmark produces highly stable performance estimates.

Table 8: Performance stability across five independent runs.

| Model | Overall Score | Memory Score |
|---|---|---|
| gpt-4o-mini | 0.3178 (±0.0025) | 0.1504 (±0.0044) |
| gpt-4.1-mini | 0.3675 (±0.0023) | 0.2030 (±0.0031) |
| gemini-2.5-flash-lite | 0.3921 (±0.0062) | 0.2583 (±0.0100) |
| gemini-2.5-flash | 0.4465 (±0.0037) | 0.3240 (±0.0056) |

