# OpenReview forum: "AMemGym: Interactive Memory Benchmarking for Assistants in Long-Horizon Conversations"
_ICLR.cc/2026/Conference — ICLR 2026 Poster_

### Official Review · Reviewer_rDSg · 2025-10-21

**Soundness:** 3
**Presentation:** 2
**Contribution:** 2
**Rating:** 6
**Confidence:** 3

**Summary:**

The paper introduces AMemGym, an interactive framework for evaluating and optimizing memory management in long-horizon conversations with LLM-based assistants. It addresses limitations of static, off-policy benchmarks by enabling on-policy evaluation through simulated users and structured data. AMemGym supports memory personalization, role-play-based latent state revelation, and structured state evolution. Experimental results highlight its capability to identify performance gaps and foster the self-evolution of memory systems, making it a scalable and diagnostic tool for conversational assistant development

**Strengths:**

1. The paper introduces a novel on-policy evaluation framework, AMemGym, which effectively addresses limitations in existing memory evaluation systems.

2. The integration of simulated users and structured state evolution ensures reliable and scalable assessments of memory capabilities.

3. Experimental evidence supports the framework’s ability to foster memory self-evolution, offering practical insights for conversational assistant optimization.

**Weaknesses:**

1. The statistical information about the data is insufficiently described. For example, what categories of dialogues are included in the dataset? How many dialogues are there in total? What is the distribution of token lengths across these dialogues? These details are missing throughout the paper and should be explicitly addressed.

2. The "Memory Implementation" section on page 6 is somewhat confusing. How exactly is the AWE method implemented? Why is parameter tuning only applied to the AWE method? Does this imply that the RAG and AWI methods do not have relevant parameters to adjust?

3. It is unclear why the paper does not adopt common generation evaluation metrics, such as GPT4Judge or BLEU, for performance assessment. Instead, it uses a self-constructed memory score for the final evaluation. The rationale behind this choice should be explained in more detail, especially given the availability of well-established evaluation metrics.

4. The structure of the appendix appears overly simplistic. It merely lists prompts without clearly explaining at which stage of the process each prompt was used. This lack of context makes the appendix difficult to interpret and detracts from its utility.

**Questions:**

See weaknesses.

---

> ### Author Response · Authors · 2025-11-21
>
> Thank you for your comprehensive review and for recognizing the novelty of our integrative grounding benchmark, the broad baseline coverage, and the actionable empirical insights we provide.
>
>
> > W(eakness) 1: Lack of dataset details
>
> A(nswer) 1:
>
> Our benchmark consists of **unique user profiles** (20 test) spanning
>   **11 distinct conversational categories** including Lifestyle & Personal (20%),
>   Professional Services (15%), Administrative (15%), and 8 additional categories.
>
>   **User diversity**: Age range 18-85 years (6 age groups), 9 education levels (from
>   incomplete high school to graduate degrees).
>
>   **Conversation structure**:
>   - base: 11 periods per user, avg 4.29 sessions per period, 47.15 total turns
>   - extra: 21 periods per user, avg 3.89 sessions per period, 81.60 total turns
>
>   **Token statistics**: User queries average ~21 tokens (range: 13-32), evaluation
>   answers average ~60 tokens (range: 39-98). Overall dialogue length varies for different models due to the on-policy interaction property, ranging from 60K ~ 140K on average.
>
>   **Evaluation complexity**: 10 questions per user requiring 2-3 memory states each,
>   with 4-7 answer choices per question, testing multi-hop memory retrieval and reasoning.
>
>  | Metric                 | Base | Extra |
>   |------------------------|----------|------------|
>   | Total Users          | 20         | 20         |
>   | Total Periods         | 220        | 420        |
>   | Total Sessions (Turns) | 943        | 1,632      |
>   | Evaluation Questions   | 200        | 200        |
>   | Avg Turns per User      | 47.15      | 81.60      |
>
> User diversity
>  - Age: 19-85 years, 6 age groups
>   - Education: 9 levels
>   - 16 distinct occupations

---

> ### Author Response · Authors · 2025-11-21
> **Official Comment by Authors (2/n)**
>
> > W2: Unclear explanation of memory methods
>
> A2:
>
> We thank the reviewer for pointing out this lack of clarity. We agree that the implementation details and the rationale for our parameter tuning strategy were not sufficiently explained. We provide detailed explanations below and will revise Section 4.1 to make these distinctions clearer.
>
> A. `How exactly is the AWE (Agentic Write, External) method implemented? `
>
> AWE operates on a **two-tier memory architecture** combining short-term context with long-term external storage. The system maintains a **sliding message buffer** (`local_msgs`) that holds recent conversation turns, while using an **external vector database** (powered by mem0/Milvus) for persistent memory storage.
>
> **Write Process (Agentic Memory Extraction):**
>
> As the agent interacts with the user, conversation messages accumulate in the local buffer. When the buffer reaches a threshold (`update_bsz + local_length`), AWE triggers memory extraction:
>
>   1. The system takes the oldest `update_bsz` messages from the buffer
>   2. These messages are passed to the mem0 library with `infer=true`, which internally uses an LLM to extract structured, factual information rather than storing raw conversation text
>   3. Extracted facts are embedded (using `text-embedding-3-small`) and stored in the vector database
>   4. The processed messages are removed from the buffer, making room for new interactions
>
> This agentic extraction is the key distinction from standard RAG: instead of storing *"I prefer 6-12 people at my bridge gatherings,"* AWE extracts and stores the structured fact `bridge_gathering_group_size: medium_6_12`.
>
> **Read Process (Embedding-Based Retrieval):**
>
> When answering a question, AWE performs semantic retrieval:
>
>   1. The user's question is embedded using the same embedding model
>   2. The system performs similarity search in the vector database to retrieve the top-k most relevant memories
>   3. Retrieved memories are formatted chronologically and injected into the system prompt:
>      "You are a helpful AI. Respond according to retrieved memories.
>      Relevant user memories ordered by time (earliest to latest):
>   - [memory 1]
>   - [memory 2]
>      ..."
>   4. The LLM generates a response using both retrieved memories and the recent conversation context (`local_msgs`)
>
>
> B. `Why is parameter tuning only applied to the AWE method?  Does this imply that the RAG and AWI methods do not have relevant parameters to adjust?`
>
> We frame AWE as the core architecture. The other methods are its simplified variants:
>
> - Standard RAG is equivalent to AWE with a non-selective, "always write" policy that stores raw conversational text.
> - AWI (Agentic Write, In-Context) is AWE without an external vector store (use a buffer area in the context window instead), bottlenecked by the LLM's context window.
>
> We concentrated our hyperparameter analysis on AWE because, as the most representative memory implementation paradigm in current memory study / projects, such as mem0 [1], A-Mem [2], and MemOS [3], it offers the richest diagnostic value.
>
> AWE's design involves critical trade-offs between write frequency (freq), short-term buffer size (ns), and retrieval count (topk).
> Analyzing these parameters allows us to demonstrate AMEMGYM's ability to diagnose complex system behaviors, as shown in Figure 7.
>
> RAG and AWI also involve non-enumerable configurations. We treat these as variations of writing strategy and keep relevant parameters—such as freq, ns, topk for RAG and freq, ns for AWI—consistent with the AWE-(2,4,30) baseline. It provides a controlled comparison along another axis of writing mechanisms.
>
>
>
> > References
> 1. Chhikara, Prateek, Dev Khant, Saket Aryan, Taranjeet Singh, and Deshraj Yadav. "Mem0: Building production-ready ai
>   agents with scalable long-term memory." arXiv preprint arXiv:2504.19413, 2025.
> 2. Xu, Wujiang, et al. "A-mem: Agentic memory for llm agents." arXiv preprint arXiv:2502.12110 (2025).
> 3. Li, Zhiyu, et al. "Memos: A memory os for ai system." arXiv preprint arXiv:2507.03724 (2025).

---

> ### Author Response · Authors · 2025-11-21
> **Official Comments by Authors (3/4)**
>
> > W3: poor justification for evaluation metric
>
> A3:
>
> We selected a multiple-choice format to ensure stable and precise evaluation, a choice consistent with leading memory benchmarks [1-4]. This approach directly tests an agent's ability to recall specific facts, avoiding the pitfalls of open-ended evaluation. Metrics like BLEU fail to capture factual accuracy, while LLM judges are often expensive, inconsistent, and unable to verify if the correct memory was used, frequently rewarding plausible but incorrect answers.
> Furthermore, our metric design provides deeper insight. The **memory score** isolates memory decay from overall task performance for clearer analysis, and our **diagnostic metric**s pinpoint the root causes of memory failures. This creates a reliable and interpretable framework for targeted system improvement.
>
>
> References
> 1. Wu, Di, Hongwei Wang, Wenhao Yu, Yuwei Zhang, Kai-Wei Chang, and Dong Yu. 2024. "Longmemeval: Benchmarking chat
>   assistants on long-term interactive memory." arXiv preprint arXiv:2410.10813.
> 2. Jiang, Bowen, Zhuoqun Hao, Young-Min Cho, Bryan Li, Yuan Yuan, Sihao Chen, Lyle Ungar, Camillo J Taylor, and Dan
>   Roth. 2025. "Know me, respond to me: Benchmarking llms for dynamic user profiling and personalized responses at scale."
>    arXiv preprint arXiv:2504.14225.
> 3. Du, Yiming, Hongru Wang, Zhengyi Zhao, Bin Liang, Baojun Wang, Wanjun Zhong, Zezhong Wang, and Kam-Fai Wong. 2024.
>   "Perltqa: A personal long-term memory dataset for memory classification, retrieval, and fusion in question answering."
>   In Proceedings of the 10th SIGHAN Workshop on Chinese Language Processing (SIGHAN-10), 152–164.
> 4. Maharana, Adyasha, Dong-Ho Lee, Sergey Tulyakov, Mohit Bansal, Francesco Barbieri, and Yuwei Fang. 2024. "Evaluating
>    very long-term conversational memory of llm agents." In Proceedings of the 62nd Annual Meeting of the Association for
>   Computational Linguistics (Volume 1: Long Papers), 13851–13870.
>
>
>
>
>
> > W4: appendix structure (prompts)
> A4:
> We add detailed descriptions for the prompts according to the workflow of the AMEMGYM framework as presented in the main body of the paper.
>
> C.1 Prompts for Structured Data Generation (corresponds to Sec 3.1):
>
> This section contains the prompts used in the initialization phase (Section 3.1) to construct the evaluation blueprint. These prompts operate offline to generate the ground-truth data before any agent interaction occurs.
>
> - User Profile and State Schema Sampling
>
> 	These prompts (Sample User Profiles, Sample User Questions, Refine State Schema) initialize the simulation. They sample a base persona from the Nemotron dataset and iteratively define a canonical schema of state variables (e.g., mentoring_delivery_format) and their possible values, ensuring the user has a consistent set of attributes to track.
>
> - User States Evolution
>
> 	These prompts (Sample Initial State, Sample State Updates, Elaborate State Updates) simulate the temporal dynamics of the user. They generate the ground-truth trajectory of state changes across periods ($T_\sigma$) and create narrative "life events" that justify why a preference or situation changed (e.g., moving houses or changing jobs).
>
> - Query Generation (for state exposure)
>
> 	These prompts (Sample Update/Initial Queries, Refine Query) bridge the gap between structured states and natural language. They generate the specific utterances ($u_{t,k}$) the simulated user will say to implicitly reveal their hidden state to the agent, ensuring the conversation is grounded in the pre-generated schema.
>
> - Personalized Answer Generation and Reflection
>
> 	These prompts (Sample Personalized Answers, Check/Refine Personalized Answer) generate the evaluation QA pairs. Crucially, they include a "reflection" step where an LLM validator ensures the generated answer corresponds strictly to the specific state variant, guaranteeing that the ground-truth labels are unambiguous.
>
>
> C.2 Prompts for On-Policy Interaction (corresponds to Sec 3.2):
>
> - User Simulator System Prompt
>
> This is the core instruction set for the User Simulator (Generate User Follow-up Prompt). It directs the LLM to role-play the specific persona, manage conversation flow, and naturally introduce the "exposure" utterances generated in C.1.

---

> ### Author Response · Authors · 2025-11-21
> **Official Comment by Authors (4/4)**
>
> C.3 Prompts for Evaluation (corresponds to Sec 3.3):
>
> This section will detail the specific prompts used for:
>
> - Overall and Utilization Evaluation
>
> 	The Overall Evaluation Prompt presents the agent with the multiple-choice question based on its memory. The Utilization Evaluation Prompt provides the agent with the ground-truth state explicitly, which isolates reasoning capabilities from retrieval capabilities to calculate the Utilization Score.
>
> - Diagnostic Evaluation
>
> 	The Agent State Diagnosis Prompt is used to calculate Write and Read failure rates. It asks the agent to explicitly state its belief regarding specific user variables (e.g., "What is the current value for 'mentoring_delivery_format'?"). This allow us to compare the agent's internal state against the ground truth.
>
>
> C.4 Prompts for Memory Self-Evolution (corresponds to Sec 5):
>
> This section includes the prompts used in the agent optimization experiments (Section 5).
>
> - Memory Policy Self-Evolution
>
> 	This prompt feeds the environmental feedback (summary of failures) into the agent's optimizer. It instructs the LLM to rewrite the "Types of Information to Remember" section of the system prompt to address specific failure modes identified in previous cycles.

---

### Official Review · Reviewer_bGms · 2025-10-23

**Soundness:** 3
**Presentation:** 3
**Contribution:** 3
**Rating:** 6
**Confidence:** 4

**Summary:**

The paper introduces **AMEMGYM**, an **interactive (on-policy)** benchmarking and optimization environment for long-horizon conversational memory. A **structured state-evolution blueprint** anchors free-form, LLM-driven role-play and enables **diagnostic** scoring with attribution to **write / read / utilization** stages. Experiments compare native LLMs, RAG, and two agentic-write variants, reveal sizable **on- vs off-policy** ranking shifts, and demonstrate feedback-driven **self-evolution** of memory policy.

**Strengths:**

- **Clear motivation**: Off-policy evaluations can induce reuse bias; AMEMGYM offers an on-policy, diagnostically rich setup.
- **Methodological novelty**: Persona/state trajectories, exposure utterances, QA variants (with reflection) enable constrained interaction and automated scoring; normalized memory score and stage-wise failure analysis are useful.
- **Thorough evaluation**: Quantifies on- vs off-policy discrepancies; characterizes long-horizon degradation of native LLMs; provides granular failure attributions and analyses of frequency, short-term buffers, and top-k.
- **Meta-evaluation**: Human ratings on exposure clarity and dialogue consistency support data quality.

**Weaknesses:**

- **External validity of simulated users**: Add a small human-in-the-loop comparison and a systematic study of user-LLM choice.
- **Broader baselines**: Include structured memory graphs/event stores, hierarchical compression, and explicit state trackers.
- **Leakage control**: Provide anti-leak prompt design and automatic leakage checks.
- **Metric reporting**: Add variance/CI and difficulty-conditioned analyses for the normalized memory score.
- **Scope of self-evolution**: Jointly evolve retrieval and utilization (e.g., top-k plus utilization prompting), and report stability/convergence.

**Questions:**

1. **Trace reusability**: Must interactions be regenerated for each system, or can exposure prompts and user policy be reused for fair comparison?
2. **Noise control**: How is the ratio of non-informative chatter quantified and manipulated as \(N_i\) grows?
3. **Upper bound (SUB)**: Does the UB solver access all ground-truth states plus a strong reasoner? Would an expert-solver UB better isolate utilization bottlenecks?
4. **Privacy**: Guidance for minimizing PII in real-user deployments?

---

> ### Author Response · Authors · 2025-11-21
>
> Thank you for your comprehensive review and for recognizing the novelty of our integrative grounding benchmark, the broad baseline coverage, and the actionable empirical insights we provide.
>
>
> > W(eakness) 1: External validity of simulated users: Add a small human-in-the-loop comparison and a systematic study of user-LLM choice.
>
> A(nswer) 1:
>
> This is an excellent point. We have conducted the experiments you suggested. We have not only performed additional studies involving different user-simulator LLMs but also completed a human-agent meta-evaluation.
>
> - Results in Appendix F.2 shows **the comparison of results achieved by using a different user simulator**, deepseek-v3. We found that the choice of user LLMs has minimal impact on the quality of evaluation.
>
> - To further answer your question, in addition to the meta-evaluation in the paper, we conduct **an additional meta-evaluation** to validate the reliability of the simulator's judgments. Specifically, we randomly sampled 100 questions from the model's evaluation logs and employed two independent annotators to label them. We then calculated the agreement rates between the two humans (inter-annotator agreement) as well as the agreement between each human and the LLM simulator. This meta-evaluation confirms that the simulator achieves near-perfect consistency with human judgment. This demonstrates that the simulator accurately reflects human-level interpretation, ensuring it serves as a reliable foundation for benchmarking.
>
> | QA Annotation Pair | Agreement |
> | :--- | :---: |
> | User1 - LLM | 0.96 |
> | User2 - LLM | 0.94 |
> | User1 - User2 | 0.92 |
>
>
>
> > W2: Broader baselines: Include structured memory graphs/event stores, hierarchical compression, and explicit state trackers.
>
> A2: Thank you for the suggestion. Our rationale for selecting the current baselines was to conduct a controlled analysis of core architectural principles (like Agentic Write) to provide clear, generalizable insights, rather than merely integrating multiple complex "black-box" systems. Our baselines (e.g., an implementation based on mem0) already represent a class of abstract memory architectures.
>
> We added results of a few other open-source memory frameworks below, with AWE in the paper also listed for a comparison.
>
> | Open-source Memory System | Overall | Memory |
> | :--- | :---: | :---: |
> | Mem0 w/o Graph (AWE in the paper) | 0.430 | 0.296 |
> | Mem0 w/ Graph | 0.424 | 0.284 |
> | A-Mem | 0.378 | 0.220 |
> | Nemori | 0.385 | 0.231 |
>
>
> References
> [1] Mem0: Chhikara, Prateek, Dev Khant, Saket Aryan, Taranjeet Singh, and Deshraj Yadav. "Mem0: Building production-ready ai
>   agents with scalable long-term memory." arXiv preprint arXiv:2504.19413, 2025.
> [2] A-Mem: Xu, Wujiang, et al. "A-mem: Agentic memory for llm agents." arXiv preprint arXiv:2502.12110 (2025).
> [3] Nemori: Nan, Jiayan, Wenquan Ma, Wenlong Wu, and Yize Chen. "Nemori: Self-organizing agent memory inspired by cognitive science." arXiv preprint arXiv:2508.03341 (2025).
>
>
>
>
> > W3: Leakage control: Provide anti-leak prompt design and automatic leakage checks.
>
> A3:
>
> We would like to clarify that there won’t be any information leakage problem because the agents are prompted with evaluation questions like as shown in Figure 2, specifically designed without any target state information. Tested agents must utilize information in its own memory based on the on-policy interactions.
>
>
>
> > W4: Metric reporting: Add variance/CI and difficulty-conditioned analyses for the normalized memory score.
> A4:
>
> 1. To assess reliability, we repeated the experiment five times across a subset of models. The mean and standard deviation values, reported below, demonstrate that our benchmark yields highly stable performance estimates.
>
> | Model | Overall Score | Memory Score |
> | :--- | :---: | :---: |
> | gpt-4o-mini | .3178 (±.0025) | .1504 (±.0044) |
> | gpt-4.1-mini | .3675 (±.0023) | .2030 (±.0031) |
> | gemini-2.5-flash-lite | .3921 (±.0062) | .2583 (±.0100) |
> | gemini-2.5-flash | .4465 (±.0037) | .3240 (±.0056) |
>
>
> 2. Difficulty-conditioned analysis: We appreciate the suggestion to make this analysis more explicit. Our experimental framework was designed with this capability in mind. In fact, the comparison between the 'base' and 'extra' configurations, which is currently in Section 4.1 and **Appendix F.1**, serves as a direct implementation of a difficulty-conditioned analysis. The 'extra' configuration represents a higher difficulty level by design. Thanks to our fully automatic data generation framework. Difficulty control can be easily accessed by tuning parameters mentioned in the paper: the number of evolution periods N_p(quantity of key information), required states per question N_s(reasoning depth), and interaction turns per state exposure N_i (noise level).

---

> ### Author Response · Authors · 2025-11-21
> **Official Comment by Authors (Cont'd)**
>
> > W5: Scope of self-evolution: Jointly evolve retrieval and utilization (e.g., top-k plus utilization prompting), and report stability/convergence.
>
> A5: We appreciate this forward-looking suggestion. We position the self-evolution section in our current paper as a proof-of-concept, intended to demonstrate the feasibility of policy optimization via our benchmark environment. The joint evolution of retrieval and utilization strategies you mentioned is an excellent direction for future work.
>
>
>
>
> # Questions
>
> > Trace reusability: Must interactions be regenerated for each system, or can exposure prompts and user policy be reused for fair comparison?
>
> A1: For a strictly on-policy evaluation, the interaction trace must be regenerated for each agent being evaluated. This is because each of the agent's responses influences the user's subsequent actions. Reusing a conversational trace would shift the evaluation paradigm to "mildly off-policy."
>
> While this has less impact on evaluating stateless "models," it fails to capture an "agent's" ability to alter the environment through interaction, which is a core tenet of our benchmark.
>
> > Noise control: How is the ratio of non-informative chatter quantified and manipulated as (N_i) grows?
>
> A2: In our framework, non-informative chatter (i.e., conversational content irrelevant to the user's state) arises naturally during the dialogue; we do not exert hard control over its ratio. The natural increase of this "noise" as the conversation length N_i grows is an intrinsic part of the benchmark, which leads to an increased amount of information not tracked by target user states. This simulates the challenge of long-horizon conversations, where the agent must sift through an increasing amount of irrelevant information to identify key facts.
>
> > Upper bound (SUB): Does the UB solver access all ground-truth states plus a strong reasoner? Would an expert-solver UB better isolate utilization bottlenecks?
>
> A3: Yes, it has access to **all relevant ground-truth states** for the current query. Its purpose is to evaluate the upper bound of the assistant model's own capabilities under the assumption of perfect memory retrieval. Therefore, using an external "expert solver" would not measure the performance ceiling of the assistant itself, which would be inconsistent with our evaluation goals.
>
> > Privacy: Guidance for minimizing PII in real-user deployments?
>
> A4: As our benchmark is based entirely on simulated users that are purely synthetically generated [1] to simulate demographic distribution, no real-user personally identifiable information (PII) is involved in the current work. We acknowledge that when such technologies are deployed in the real world, privacy protection is paramount. In such cases, strict techniques like data anonymization, encryption, and access control would be essential, but this falls outside the scope of our current research.
>
> > Reference(s)
>
> [1] https://huggingface.co/datasets/nvidia/Nemotron-Personas-USA

---

> > ### Comment · Reviewer_bGms · 2025-11-24
> > **comments**
> >
> > I appreciate the authors’ rebuttal.
> > The response is helpful, but I decide to keep my initial score.

---

### Official Review · Reviewer_fnRa · 2025-10-30

**Soundness:** 4
**Presentation:** 4
**Contribution:** 3
**Rating:** 8
**Confidence:** 3

**Summary:**

The work introduces an interactive environment designed to benchmark the memory capabilities of llm assistants. The authors argue that existing benchmarks are flawed because they rely on static, off-policy data. This means the assistant is evaluated on a fixed conversation history it did not create, which fails to capture how an assistant's own responses influence the dialogue and can lead to unreliable evaluations. They first generate the state of the user using structured outputs and then the simulated user uses their attributes in a natural way during the conversation. The framework also introduces diagnostic metrics that decompose memory failures into three stages: write, read, and utilization. They demonstrate that this "gym" can be used for agent self-evolution, where an agent uses the environment's feedback to autonomously improve its own memory-writing policy

**Strengths:**

1. The papers central critique of off-policy evaluation is compelling and well-articulated. The authors provide concrete evidence in table 2 that evaluation rankings of memory systems change when moving from an off-policy to an on-policy setup, proving that the distinction is not just theoretical but has practical consequences.
2. The introduction of write, read, and utilization failure metrics which gives better insights into failure modes than the usual accuracy metric.
3. The self-evolution experiment shows that agents can improve its memory policy by learning from the environment's feedback.

**Weaknesses:**

1. The work only evaluates memory for selecting the correct answer using multiple choice questions, but doesn't test the generation capabilities.
2. The memory is tested using structured key-value pairs and doesn't test the episodic memory or memory where the assistant has to reason over multiple facts.

**Questions:**

1. In section 5, "Complete Feedback" is described as including the questions, agent's answers, and ground-truth answers, which are summarized into <feedback.summary>. Could you provide an example of how this summary is formatted? Is it a natural language paragraph, structured JSON, or another format?
2. How do you ensure diversity in the structured generation?

- Some weird formatting: L373-375

---

> ### Author Response · Authors · 2025-11-21
>
> Thank you for your comprehensive review and for recognizing the novelty of our integrative grounding benchmark, the broad baseline coverage, and the actionable empirical insights we provide.
>
>
> > W(eakness) 1: Limited evaluation scope focusing on multiple-choice questions, not generative capabilities.
>
> A(nswer) 1:
>
> This was a deliberate design choice to ensure evaluation robustness. Automated evaluation of free-form generation is prone to noise and evaluator variance, which could obscure the specific memory-related failures we aim to isolate. The multiple-choice format provides a clean and objective measure of memory performance. **This choice aligns with contemporary memory evaluation benchmarks** including LongMemEval [1], PersonaMem [2], PerLTQA [3], and LoCoMo [4], all of which employ QA accuracy as a primary metric for memory assessment.
>
> Leveraging the grounded structured data paired with the interactions allows us to bypass the uncertainty introduced by LLM-based judges, thereby ensuring reliable and robust evaluation outcomes.
>
>
> > W2: Memory evaluation is limited to structured key-value pairs, neglecting episodic memory and multi-fact reasoning.
>
> A2:
>
> We appreciate the opportunity to clarify this. Our benchmark is **designed specifically to test episodic and semantic memory** -- the ability to recall facts and events from the ongoing conversation across different periods. The tested agents must process both episodic experiences (what occurred during the growing on-policy interactions) and semantic facts (what the long-term conversation reveals about the user) to demonstrate effective memory. The key-value pairs are intentionally hidden from the agents and serve solely as a ground-truth representation, ensuring a robust and trustworthy evaluation framework.
>
> The benchmark also incorporates multi-fact reasoning in two key ways:
>
> 1. Fact integration: Agents need to recall and utilize multiple facts learned at different positions in the conversation. (1 question requires multiple key information)
>
> 2. State evolution: User preferences can change over the dialogue (e.g., "I used to like coffee, but now I prefer tea"). Correctly answering requires the agent to track this evolution between related facts. This is a form of temporal reasoning.
>
> > References
>
> 1. Wu, Di, Hongwei Wang, Wenhao Yu, Yuwei Zhang, Kai-Wei Chang, and Dong Yu. 2024. "Longmemeval: Benchmarking chat  assistants on long-term interactive memory." arXiv preprint arXiv:2410.10813.
> 2. Jiang, Bowen, Zhuoqun Hao, Young-Min Cho, Bryan Li, Yuan Yuan, Sihao Chen, Lyle Ungar, Camillo J Taylor, and Dan Roth. 2025. "Know me, respond to me: Benchmarking llms for dynamic user profiling and personalized responses at scale." arXiv preprint arXiv:2504.14225.
> 3. Du, Yiming, Hongru Wang, Zhengyi Zhao, Bin Liang, Baojun Wang, Wanjun Zhong, Zezhong Wang, and Kam-Fai Wong. 2024. "Perltqa: A personal long-term memory dataset for memory classification, retrieval, and fusion in question answering." In Proceedings of the 10th SIGHAN Workshop on Chinese Language Processing (SIGHAN-10), 152–164.
> 4. Maharana, Adyasha, Dong-Ho Lee, Sergey Tulyakov, Mohit Bansal, Francesco Barbieri, and Yuwei Fang. 2024. "Evaluating very long-term conversational memory of llm agents." In Proceedings of the 62nd Annual Meeting of the Association for Computational Linguistics (Volume 1: Long Papers), 13851–13870.

---

> ### Author Response · Authors · 2025-11-21
> **Official Comment by Authors (Cont'd)**
>
> # Questions:
>
> > Q1: Could you provide an example of how the <feedback.summary> is formatted?
>
> Certainly. The `<feedback.summary>` in our self-evolution framework (Section 5) is a JSON-formatted structure containing evaluation results from a period. Here is a concrete example from our implementation:
>
> **Example (Complete Feedback with Answer):**
>
>   ```json
>   {
>     "question_answer_history": [
>       {
>         "question": "Question: What are some engaging activities I can organize for my monthly bridge
>   gatherings to keep them fresh and enjoyable?;\n(A) For a group of experienced bridge enthusiasts, try
>    rotating partnerships each round...;\n(B) With a larger and diverse crowd, consider organizing a
>   mini-tournament...;\n(C) For a small, close-knit gathering of seasoned players, focus on relaxed
>   play...;\n(D) In a moderately sized group of older adults, set up duplicate bridge sessions...;\n(E)
>   With a mix of ages and a moderate group size, try pairing experienced players with younger ones...;",
>         "assistant_response": "A",
>         "ground_truth": "E",
>         "retrieved_memories": [
>           "bridge_gathering_group_size: medium_6_12",
>           "bridge_gathering_guest_age_range: mostly_50_plus"
>         ]
>       },
>       {
>         "question": "Question: How can I best mentor young women in my community to support their
>   personal and professional growth?;\n(A) Organize a series of interactive workshops...;\n(B) Design a
>   workshop series focused on effective teaching strategies...;\n(C) Facilitate small group
>   sessions...;",
>         "assistant_response": "B",
>         "ground_truth": "A",
>         "retrieved_memories": [
>           "mentoring_focus_area_for_young_women: community_leadership",
>           "mentoring_delivery_format: small_group"
>         ]
>       }
>     ],
>     "user_information_updates": {
>       "bridge_gathering_guest_age_range": "mixed_ages_with_young_adults",
>       "mentoring_delivery_format": "workshop_series",
>       "rose_garden_maintenance_frequency": "monthly_minimal"
>     }
>   }
> ```
>
> Structure Explanation:
>   - question_answer_history: List of evaluation questions with:
>     - question: Formatted question with multiple-choice options (A-E)
>     - assistant_response: Agent's selected answer (e.g., "A")
>     - ground_truth: Correct answer based on user's actual state (e.g., "E")
>     - retrieved_memories: Memories the agent retrieved from its memory system when answering
>
>   - user_information_updates: State changes revealed during the period's conversations
>
>
> > Q2: How do you ensure diversity in the structured generation?
>
> Thank you for this excellent question. Ensuring diversity in the structured data generation is a core design principle of AMEMGYM, and we achieve this through a multi-layered approach, as detailed in Section 3.1.
>
> 1. Diverse user profiles as foundation: We start by sampling from a large and varied pool of user profiles. Our experiments use the Nemotron-Personas dataset (lines 158-161), which contains 100,000 personas grounded in diverse demographic and personality distributions, ensuring a broad foundation for our simulations.
>
> 2. Diverse scenarios: For each profile, our generation prompts (Appendix C.1) are explicitly designed to create variety. They generate diverse questions covering "both user-specific and general life topics" and simulate diverse state evolution trajectories by encouraging changes to be "spread across different variables" over time (lines 164-180).
>
>
>
> > Other comments:
>
> Thank you for catching the formatting issue on L373-375.

---

### Official Review · Reviewer_2oDj · 2025-11-03

**Soundness:** 2
**Presentation:** 3
**Contribution:** 2
**Rating:** 4
**Confidence:** 4

**Summary:**

The paper introduces AMEMGYM, a new interactive, on-policy benchmark designed to  evaluate memory management in long-horizon conversational assistants. AMEMGYM differs from existing static datasets and provides a fully automated environment where LLM-simulated users engage in structured evolving conversations that reveal latent user states through role-play. Through experimental comparisons, they found that on-policy evaluation changed the rankings and scores substantially. They also show that even the latest LLMs struggle to maintain long-term conversational memory, confirming context-length degradation. Agentic-Write External systems performed best which highlights that selective, structured writing yields higher information utilization accuracy.

**Strengths:**

* The paper is grounded on a novel motivation: existing benchmarks mostly focus on off-policy memory evaluation which might have a gap between realistic systems.
* The introduction of write/read/utilization decomposition is a meaningful contribution.
* The paper also conducted extensive comparisons that shed light on long-context degradation and memory design trade-offs.

**Weaknesses:**

* While the paper positions on-policy evaluation as a core contribution, the empirical and conceptual justification for its advantage over off-policy settings remains underdeveloped. Although Table 2 and Figure 5 show some rank changes between on- and off-policy settings, the paper does not provide insights on why those differences matter. It remains unclear what specific behavioral aspects of “interactive memory” are uniquely captured. In fact, one could argue that well-curated off-policy datasets offer simpler, cheaper, and more reproducible alternatives, and the paper does not convincingly rule out this possibility.
* The claimed novelty of introducing an on-policy memory evaluation environment is somewhat weakened by the existence of several recent long-horizon, on-policy frameworks: AgentGym, DeepResearch, SWE-Agent. These already feature real-time decision-making, persistent contexts, and memory management. The paper should sufficiently clarify what AMEMGYM contributes beyond these environments.
* The entire benchmark relies on LLM-simulated users rather than real human interaction data. While this enables scale and control, it raises a question: whether simulated dialogues genuinely capture the noise, inconsistency, and ambiguity of real users. Even though the authors conduct a “meta-evaluation”, it mostly checks internal coherence rather than human-likeness or behavioral realism.
* Although the paper evaluates several general architectures (e.g. RAG, agentic write, and long-context LLMs), it does not include direct experiments on established open-source memory frameworks such as Mem0 (Chhikara et al., 2025) or A-Mem (Xu et al., 2025), despite citing both as prior work. While the authors’ custom agentic write setups are reasonable for controlled analysis, the paper should better justify why this self-defined configuration was chosen instead of real implementations that are already widely used.

**Questions:**

* Could you provide the cost analysis?
* Could you also provide results on popular open-source models?

---

> ### Author Response · Authors · 2025-11-21
>
> Thank you for your comprehensive review and for recognizing the novelty of our integrative grounding benchmark, the broad baseline coverage, and the actionable empirical insights we provide.
>
>
> > W(eakness) 1: Underdeveloped justification for on-policy evaluation
>
> A(nswer) 1:
>
> We appreciate your suggestion for a more structured framing.
>
> The key advantage of on-policy evaluation lies in its ability to capture the **realistic assistant–user interaction loop and its temporal dynamics**, aspects that static, off-policy benchmarks inherently overlook. Memory assistants iteratively update their internal states based on prior interactions and adapt their behavior in subsequent turns. Off-policy datasets, by relying on external interactions, can introduce biases in memory updates that propagate across turns and amplify over long conversations, leading to less reliable feedback for system selection or optimization.
>
> For instance, a memory system that fails to memorize a fact such as "the user broke his leg in an accident" will generate different responses in later, related interactions—for example, to a query like "any recommended activities to relax?". Furthermore, memory systems may reinforce related memories through their responses. Off-policy evaluation, being based on static interactions, cannot capture these dynamics and recurrent dependencies.
>
> We acknowledge that a well-curated static benchmark could serve as a more economical proxy for on-policy evaluation if its equivalence can be rigorously validated. Our work highlights the evaluation biases associated with off-policy reuse—issues that, to our knowledge, have not been sufficiently addressed in prior memory evaluation research and may have misdirected system development. Exploring such a proxy remains valuable future work for both us and the community.
>
>
>
> > W2: The claimed novelty of introducing an on-policy memory evaluation environment is somewhat weakened by the existence of several recent long-horizon, on-policy frameworks:
>
> A2:
>
> Yes, the growing focus on environment-based, on-policy frameworks is a clear and welcome trend, as they are essential for evaluating LLM agents in realistic, long-horizon scenarios. This trend directly motivates our work. We noticed **a critical gap**: while the field moves toward interactive evaluation, the memory systems for long conversations—a fundamental agent capability—are still predominantly assessed using static, off-policy methods.
>
> Our key contribution is to resolve this discrepancy. We ask: **do these static, off-policy evaluations deliver the same results as a dynamic, on-policy test?** To answer this, AMemGym tackles the **scalability challenge of long user interactions** by leveraging a dual approach: a simulated user LLM for scalability, grounded by curated hierarchical structured data for reliability.
>
> Beyond the shared principle of environment-based evaluation, AMemGym has a distinct focus:
>
> Core task (memory in long conversations): We concentrate on the fundamental task of long conversation, which has the broadest applicability for LLM-based agents. Our framework is specifically designed to scale to extremely long interactions, emphasizing the management of episodic and semantic memory.
> Specific challenge (scaling user interactions): We pioneer the on-policy evaluation of memory strategies for understanding latent user states—a critical capability that, despite its importance, lacks a dedicated interactive evaluation framework due to the scalability challenge. Our contribution is proving this complex problem can be automated and scaled effectively using our user LLM and grounded data methodology.
>
> > W3: it raises a question: whether simulated dialogues genuinely capture the noise, inconsistency, and ambiguity of real users
>
> A3:
>
> This is a valid point regarding the trade-offs of simulation. Using real humans for on-policy evaluation at this scale is intractable due to cost and reproducibility constraints. Our approach provides a scalable, controllable, and reproducible environment.
>
> To further answer your question, in addition to the meta-evaluation in the paper, we conduct an additional meta-evaluation to validate the reliability of the simulator's judgments. Specifically, we randomly sampled 100 questions from the model's evaluation logs and employed two independent annotators to label them. We then calculated the agreement rates between the two humans (inter-annotator agreement) as well as the agreement between each human and the golden choices generated by the LLM during offline structured data construction. This meta-evaluation confirms that the simulator achieves near-perfect alignment with human judgment, which ensures it serves as a reliable foundation for benchmarking.
>
> | QA Annotation Pair | Agreement |
> | :--- | :---: |
> | User1 - LLM | 0.96 |
> | User2 - LLM | 0.94 |
> | User1 - User2 | 0.92 |

---

> ### Author Response · Authors · 2025-11-21
> **Official Comment by Authors (Cont'd)**
>
> > W4: Lack of comparison with established open-source memory frameworks
>
> A4:
>
> We apologize for the confusion. Our Agentic Write (AWE) and RAG systems are **implemented using the open-source mem0 library** (with different configurations), as noted in Footnote 2. We chose to build specific configurations (e.g., AWE-(2,4,30)) to perform a controlled analysis of core architectural principles and hyperparameters, rather than testing a single "black-box" implementation. This allows us to provide more generalizable insights (as shown in Sec. 4.3 & 4.4). Also, we added results of a few other open-source memory frameworks below, with AWE in the paper also listed for a comparison.
>
> | Open-source Memory System | Overall | Memory |
> | :--- | :---: | :---: |
> | Mem0 w/o Graph (AWE in the paper) | 0.430 | 0.296 |
> | Mem0 w/ Graph | 0.424 | 0.284 |
> | A-Mem | 0.378 | 0.220 |
> | Nemori | 0.385 | 0.231 |
>
>
> > References:
>
> 1. Mem0: Chhikara, Prateek, Dev Khant, Saket Aryan, Taranjeet Singh, and Deshraj Yadav. "Mem0: Building production-ready ai
>   agents with scalable long-term memory." arXiv preprint arXiv:2504.19413, 2025.
> 2. A-Mem: Xu, Wujiang, et al. "A-mem: Agentic memory for llm agents." arXiv preprint arXiv:2502.12110 (2025).
> 3. Nemori: Nan, Jiayan, Wenquan Ma, Wenlong Wu, and Yize Chen. "Nemori: Self-organizing agent memory inspired by cognitive science." arXiv preprint arXiv:2508.03341 (2025).
>
>
> # Questions:
>
> > Q1: Could you provide the cost analysis?
>
> Certainly. We have broken down the cost analysis into two primary components: (1) the cost of offline structured data generation per instance, and (2) the cost associated with the user-LLM for on-policy evaluation.
>
> Data Synthesis Cost: Generating the complete set of offline structured data—including questions, answer choices, and state evolution from a user profile—requires approximately 0.14M input tokens and 15.2K output tokens. Using gpt-4.1 for this construction amounts to a cost of $0.40 per instance. This minimal expense underscores the scalability of our fully automatic data construction pipeline for both evaluation and optimization purposes.
>
> User-Simulator LLM Cost: This represents the extra cost of our on-policy evaluation compared to conventional off-policy methods. Each instance requires approximately 74.5K input tokens and 2.7K output tokens for the user-LLM. This translates to a cost of $0.17 when using gpt-4.1, or just $0.02 when using deepseek-v3 (results in Appendix F.2 indicate that switching user-simulator LLMs has a minimal impact on evaluation outcomes). Critically, this additional cost for on-policy evaluation is negligible when compared to the inference cost of the LLMs being evaluated (for example, approximately $13.0 for evaluating gpt-4.1 itself).
>
>
> > Q2: Could you also provide results on popular open-source models?
>
> In addition to the results in the paper, we have included an evaluation of several leading open-source models below, with the original Gemini-2.5-Flash results provided for comparison. We are committed to tracking the latest open-source models and will maintain a public leaderboard upon the open-sourcing of this project.
>
> | Model | Overall | Memory |
> | :--- | :---: | :---: |
> | Gemini-2.5-Flash | 0.448 | 0.327 |
> | Qwen3-235B-A22B-Instruct-2507 | 0.331 | 0.148 |
> | DeepSeek-V3.1-Terminus | 0.327 | 0.151 |
> | Kimi-K2-Instruct | 0.389 | 0.234 |
> | GLM-4.6 | 0.404 | 0.257 |

---

### Author Response · Authors · 2025-11-23
**Paper Revision Note**

We sincerely thank all reviewers for their constructive feedback and insightful suggestions. We have carefully addressed each concern raised and made revisions to the manuscript. Below is a summary of the key changes:


1. **Additional meta-evaluation** (Reviewers 2oDj, rDSg): We have expanded the meta-evaluation analysis with additional details in both the main text (Section 3.4) and Appendix D.

2. **Evaluation with open-source memory systems** (Reviewers bGms, 2oDj): We have included comprehensive comparisons with other open-source memory implementations (Mem0, A-Mem, Nemori) in Appendix F.4.

3. **Expanded open-source model results** (Reviewer 2oDj): We have added evaluation results for several leading open-source models including Qwen3, DeepSeek-V3.1, Kimi-K2, and GLM-4.6 in Appendix F.5.

4. **Cost analysis** (Reviewer 2oDj): We have provided a comprehensive breakdown of computational costs and resource requirements in Appendix C.2, enabling researchers to better assess the practical feasibility of our approach.

5. **Performance stability analysis** (Reviewer bGms): We have included a reliability evaluation with mean and standard deviation across 5 independent runs for representative models in Appendix F.6, which demonstrates the benchmark's consistency.

6. **Feedback format example** (Reviewer fnRa): We have added a concrete example showing the specific format of environmental feedback used in our self-evolution framework in Appendix C.7.

7. **Benchmark statistics** (Reviewer rDSg): We have expanded the description of benchmark statistics in Section 4.1 and provided detailed breakdowns in Appendix C.1.

8. **Enhanced prompt documentation** (Reviewer rDSg): We have added more contextual descriptions for all prompts in Appendix C to better align with the paper's narrative flow and improve reproducibility.

---

### Meta-Review · Area_Chair_CP8F · 2026-01-07

**Summary:**

The authors’ rebuttal is largely responsive and supported by concrete revisions. They include additional human evaluations, provide a detailed cost analysis, broaden baseline coverage (with explicit comparisons to Mem0, A-Mem, and Nemori, and clarification that AWE is implemented using Mem0), add results for additional open-source models, present clearer dataset statistics, supply a concrete JSON example for self-evolution feedback, and report stability results across multiple runs.

The primary concerns raised in the original reviews centered on the justification for on-policy evaluation and the validity of simulated users. In response, the authors provide stronger conceptual motivation for on-policy evaluation and add a new human agreement study, which helps mitigate concerns about the reliability of the simulated user setup.

Several reviewers also noted issues with presentation clarity, particularly regarding dataset details and baseline implementations. While the rebuttal addresses these points in detail, the authors should ensure that these clarifications are fully incorporated into the main paper to improve readability and self-containment.

One remaining concern that is not fully resolved is the limited evaluation scope: the benchmark relies exclusively on multiple-choice questions. Although the authors justify this design choice for evaluation robustness, this limitation should be more explicitly acknowledged and discussed in the paper as a constraint of the current benchmark.

**Reviewer Concerns:**

1. Justification for on-policy evaluation
2. Human-in-the-loop evaluation
3. Novelty relative to existing on-policy agent frameworks
4. Other open-source memory baselines
5. Insufficient dataset statistics and prompt clarity
6. Evaluation stability and variance

**Reviewer Scores:**

Most reviewers (fnRa, bGms, rDSg) were asking clarification questions, such as statistical details of the dataset, used prompts, and other implementation details. All of these questions have been well answered by the authors, and these reviewers should either keep their current scores (which are already supportive) or increase their scores.

The reviewer 2oDj raised some major concerns about the validity of the benchmark, especially a comparison between their simulated humans and real human users, and additional baselines. The authors have also addressed these concerns nicely with new human-agreement evaluations and new baselines. I think the reviewer 2oDj would also increase their scores.

---

### Decision · Program_Chairs · 2026-01-26

Accept (Poster)